# An efficient nonconvex reformulation of stagewise convex optimization problems

**Rudy Bunel**[*]
DeepMind
rbunel@google.com

**Oliver Hinder**[*]
Google Research, University of Pittsburgh
ohinder@pitt.edu

**Srinadh Bhojanapalli**
Google Research
bsrinadh@google.com

**Krishnamurthy (Dj) Dvijotham**
DeepMind
dvij@google.com

## Abstract

Convex optimization problems with staged structure appear in several contexts, including optimal control, verification of deep neural networks, and isotonic regression. Off-the-shelf solvers can solve these problems but may scale poorly. We develop a nonconvex reformulation designed to exploit this staged structure. Our reformulation has only simple bound constraints, enabling solution via projected gradient methods and their accelerated variants. The method automatically generates a sequence of primal and dual feasible solutions to the original convex problem, making optimality certification easy. We establish theoretical properties of the nonconvex formulation, showing that it is (almost) free of spurious local minima and has the same global optimum as the convex problem. We modify PGD to avoid spurious local minimizers so it always converges to the global minimizer. For neural network verification, our approach obtains small duality gaps in only a few gradient steps. Consequently, it can quickly solve large-scale verification problems faster than both off-the-shelf and specialized solvers.

## 1   Introduction

This paper studies efficient algorithms for a particular class of stage-wise optimization problems:

$$\underset{(s,z)\in S\times \mathbf{R}^n}{\text{minimize}}\ f(s,z) \tag{1a}$$

$$\text{s.t. } \mu_i(s, z_{1:i-1}) \le z_i \le \eta_i(s, z_{1:i-1}) \qquad\qquad \forall i \in \{1, \dots, n\} \tag{1b}$$

where $n$ and $m$ are positive integers, $S \subseteq \mathbf{R}^m$, the function $f$ has domain $S \times \mathbf{R}^n$ and range $\mathbf{R}$, the functions $\mu_i$ and $\eta_i$ have domain $S \times \mathbf{R}^{i-1}$ and range $\mathbf{R}$. Given a vector $z$, we use the notation $z_{1:i}$ to denote the vector $[z_1, \dots, z_i]$. We let $z_{1:0}$ be a vector of length zero. Throughout the paper we assume that $\eta_1, \dots, \eta_n$ are proper concave functions, $f, \mu_1, \dots, \mu_n$ are proper convex functions, and $S$ is a nonempty convex set.

Problems that fall into this problem class are ubiquitous. They appear in optimal control [1], finite horizon Markov decision processes with cost function controlled by an adversary [2], generalized Isotonic regression [3, 4], and verification of neural networks [5–7]. Details explaining how these problems can be written in the form of (1) are given in Appendix A. Here we briefly outline how neural network verification falls into (1b). Letting $s$ represent the input image and $z$ the activation values, neural networks verification can be written (unconventionally) as

$$\underset{(s,z)\in S\times \mathbf{R}^n}{\text{minimize}}\ f(s,z) \text{ s.t. } z_i = \sigma([s, z_{1:i-1}] \cdot w_i),$$

---

[*]Equal contribution

for (sparse) weight vectors $w_i$ and activation function $\sigma$. A convex relaxation is created by picking functions satisfying $\mu_i(s, z_{1:i-1}) \le \sigma_i([s, z_{1:i-1}] \cdot w_i) \le \eta_i(s, z_{1:i-1})$ for all $s$ and $z$ feasible to the original problem. Solving these convex relaxations with traditional methods can be time consuming. For example, Salman et al. [8] reports spending 22 CPU years to solve problems of this type in order to evaluate the tightness of their proposed relaxation. Consequently, methods for solving these relaxations faster are valuable.

## 1.1 Related work

### 1.1.1 Drawbacks of standard solvers for stagewise convex problems

Standard techniques for solving (1) can be split into two types: first-order methods and second-order methods. These techniques do not exploit this stage-wise structure, and so they face limitations.

**First-order methods:** Methods such as mirror prox [9], primal-dual hybrid gradient (PDHG) [10], augmented lagrangian methods [11], and subgradient methods [12] have cheap iterations (i.e., a matrix-vector multiply) but may require many iterations to converge. For example,

$$\underset{x}{\text{minimize}} - x_n \quad \text{s.t.} \quad x_1 \in [0, 1], \quad -1 \le x_i \le x_{i-1} \quad \forall i \in \{1, \ldots, n-1\} \qquad (2)$$

is an instance of (1) with optimal solution at $x = \mathbf{1}$. However, this is the type of problem that exhibits the worst-case performance of a first-order method. In particular, one can show (see Appendix B) using the techniques of Nesterov [13, Section 2.1.2] it will take at least $n - 1$ iterations until methods such as PDHG or mirror-prox obtain an iterate with $x_1 > 0$ starting from $x = \mathbf{0}$. Furthermore, existing first-order methods are unable to generate a sequence of primal feasible solutions. This makes constructing duality gaps challenging. We could eliminate these constraints using a projection operator, but in general this will require calling a second-order method at each iteration, making iterations more expensive.

**Second-order methods:** Methods such as interior point and simplex methods rely on factorizing a linear system, and can suffer from speed and memory problems on large-scale problems if the sparsity pattern is not amenable to factorization. This issue, for example, occurs in the verification of neural networks as dense layers force dense factorizations.

### 1.1.2 Other nonconvex reformulations of convex problems

Most research on nonconvex reformulations of convex problems is for semi-definite programs [14–16]. In this work, the semi-definite variable is rewritten as the sum of low rank terms, forgoing convexity but avoiding storing the full semi-definite variable. Compared with this line of research our technique is unique for several reasons. Firstly, our primary motivation is speed of convergence and obtaining certificates of optimality, rather than reducing memory or iteration cost. Secondly, the landscape of our nonconvex reformulation is different. For example, it contains spurious local minimizers (as opposed to saddle points) which we avoid via careful algorithm design.

## 2 A nonconvex reformulation of stagewise convex problems

We now present the main technical contribution of this paper, i.e., a nonconvex reformulation of the stagewise convex problems of the form (1) and an analysis of efficient projected gradient algorithms applied to this formulation.

### 2.1 Assumptions

We begin by specifying assumptions we make on the objective and constraint functions in (1). Prior to doing so, it will be useful to introduce the notion of a smooth function:

**Definition 1.** *A function $h : X \to \mathbf{R}$ is smooth if $\boldsymbol{\nabla} h(x)$ exists and is continuous for all $x \in X$; $h$ is $L$-smooth if $\|\boldsymbol{\nabla} h(x) - \boldsymbol{\nabla} h(x')\|_2 \le L\|x - x'\|_2, \forall x, x' \in X$.*

**Assumption 1.** *Assume $f, \eta_1, \ldots, \eta_n, \mu_1, \ldots, \mu_n$ are smooth functions.*

**Remark 1.** *If Assumption 1 fails to hold it is may be possible to approximate $f, \eta_i$ and $\mu_i$ by smooth functions [17]. It is also possible one could use a nonsmooth optimization method [18]. However, we leave the study of these approaches to future work.*

Let $\Pi_S$ denote the projection operator onto the set $S$. Ideally, the cost of this projection is cheap (e.g., $S$ is formed by simple bound constraints) as we will be running projected gradient descent (PGD) and therefore routinely using projections.

**Assumption 2.** *Assume $S$ is a bounded set with diameter $D_s = \sup_{s,\hat{s} \in S} \|s - \hat{s}\|_2$. Further assume $Z$ is a bounded set such that for every feasible solution $(s, z)$ to (1) we have $z \in Z$. Define $D_z = \sup_{z,\hat{z} \in Z} \|\hat{z} - z\|_2$.*

We remark that if $\eta$ and $\mu$ are smooth, and $S$ is bounded then there exists a set $Z$ satisfying Assumption 2. The primary reason for Assumption 2 is it will allow us to form lower bounds on the optimal solution to (1). We will also find it useful to be able to readily construct upper bounds, i.e., feasible solutions to (1). This is captured by the following assumption.

**Assumption 3.** *For all $i \in \{1, \ldots, n\}$, if $s \in S$ and $\mu_j(s, z_{1:j-1}) \leq z_j \leq \eta_j(s, z_{1:j-1})$ for $j \in \{1, \ldots, i-1\}$ then $\mu_i(s, z_{1:i-1}) \leq \eta_i(s, z_{1:i-1})$.*

Assumption 3 is equivalent to stating that feasible solutions to (1) can be constructed inductively. In particular, given we have a feasible solution to the first $1, \ldots, i-1$ constraints we can find a feasible solution for the $i$th constraint by picking any $z_i \in [\mu_i(s, z_{1:i-1}), \eta_i(s, z_{1:i-1})]$ which must be a nonempty set by Assumption 3. All examples discussed in Appendix A satisfy Assumption 3.

## 2.2 A nonconvex reformulation

Our idea is to apply PGD to the following nonconvex reformulation of (1),

$$\underset{(s,z,\theta) \in S \times \mathbf{R}^n \times [0,1]^n}{\text{minimize}} f(s, z) \tag{3a}$$

$$\text{s.t. } z_i = (1 - \theta_i)\mu_i(s, z_{1:i-1}) + \theta_i \eta_i(s, z_{1:i-1}) \qquad \forall i \in \{1, \ldots, n\}. \tag{3b}$$

The basis of this reformulation is that if $\mu_i(s, z_{1:i-1}) \leq z_i \leq \eta_i(s, z_{1:i-1})$ then $z_i$ is a convex combination of $\mu_i(s, z_{1:i-1})$ and $\eta_i(s, z_{1:i-1})$. This reformulation allows us to replace the $z$ variables with $\theta$ variables and replaces the constraints (1b) that are difficult to project onto with box constraints. For conciseness we denote (3b) by

$$z \leftarrow \text{FORWARD}(s, \theta).$$

Let us consider an alternative interpretation of (3) that explicitly replaces $z$ with $\theta$. Define $\psi_n(s, z) := f(s, z)$ and recursively define $\psi_i$ for all $i \in \{1, \ldots, n\}$ by

$$\psi_{i-1}(s, z_{1:i-1}, \theta_{i:n}) := \psi_i(s, z_{1:i-1}, (1 - \theta_i)\mu_i(s, z_{1:i-1}) + \theta_i \eta_i(s, z_{1:i-1}), \theta_{i+1:n}).$$

Note that $\psi_{i-1}$ eliminates the variable $z_i$ from $\psi_i$ by replacing it with $(1 - \theta_i)\mu_i(s, z_{1:i-1}) + \theta_i \eta_i(s, z_{1:i-1})$. Using this notation, the reformulation (3) is equivalent to:

$$\underset{(s,\theta) \in S \times [0,1]^n}{\text{minimize}} \psi_0(s, \theta). \tag{4}$$

For intuition consider the following example

$$S := [-1, 1], \quad f(s_1, z_1) := z_1, \quad \eta_1(s_1) := 1 - s_1^2, \quad \mu_1(s_1) := s_1^2 - 1. \tag{5}$$

In Figure 1 we plot this example. Consider an arbitrary feasible point, e.g., $z_1 = 0.0$, $s_1 = 0.5$ and note that point can be written as a convex combination of a point on $\eta$ and a point on $\mu$. The nonconvex reformulation does this explicitly with box constraints replacing nonlinear constraints.

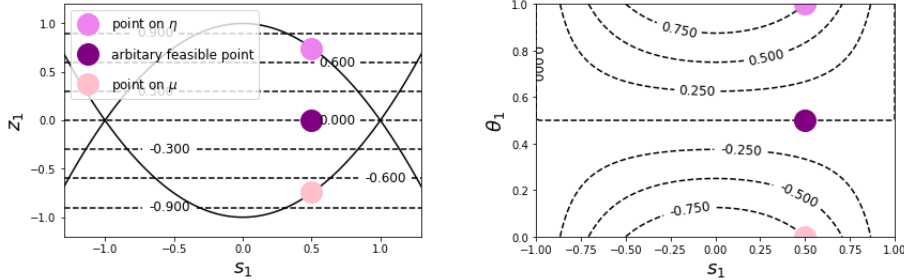

Plot of original convex problem          Plot of nonconvex reformulation $\psi_0(s_1, \theta_1)$

Figure 1: Comparison between original problem and reformulation.

The function $\psi_0$ is smooth (since it is the composition of smooth functions), and its gradient is computable by backpropagation, i.e., $\boldsymbol{\nabla}\psi_n = \boldsymbol{\nabla} f$ and for $i = n, \ldots, 1$,

$$\boldsymbol{\nabla}_s\psi_{i-1} = \boldsymbol{\nabla}_s\psi_i + \frac{\partial\psi_i}{\partial z_i}\left(\theta_i\boldsymbol{\nabla}_s\eta_i + (1-\theta_i)\boldsymbol{\nabla}_s\mu_i\right) \tag{6a}$$

$$\frac{\partial\psi_{i-1}}{\partial z_j} = \frac{\partial\psi_i}{\partial z_j} + \frac{\partial\psi_i}{\partial z_i}\left(\theta_i\frac{\partial\eta_i}{\partial z_j} + (1-\theta_i)\frac{\partial\mu_i}{\partial z_j}\right) \quad \forall j \in \{1, \ldots, i-1\} \tag{6b}$$

$$\frac{\partial\psi_0}{\partial\theta_i} = \frac{\partial\psi_i}{\partial\theta_i} = \frac{\partial\psi_i}{\partial z_i}\frac{\partial z_i}{\partial\theta_i} = \frac{\partial\psi_i}{\partial z_i}(\eta_i - \mu_i) \tag{6c}$$

where we denote $f = f(s, z)$, $\psi_i = \psi_i(s, z_{1:i-1}, \theta_{i:n})$, $\eta_i = \eta_i(s, z_{1:i-1})$, and $\mu_i = \mu_i(s, z_{1:i-1})$; this abuse of notation, where we assume these functions are evaluated at $(s, z, \theta)$ unless specified otherwise, will continue throughout the paper for the purposes of brevity. The subscript on $\boldsymbol{\nabla}$ specifies the arguments the derivative is with respect to, if it is left blank then we take the derivatives with respect to all arguments. Therefore, one can apply PGD, or other related descent algorithm to minimize $\psi_0$. Moreover, the cost of computing the gradient via backpropagation is cheap (dominated by the cost of evaluating $\boldsymbol{\nabla} f$, $\boldsymbol{\nabla}\eta$, and $\boldsymbol{\nabla}\mu$). However, since $\psi_0$ is nonconvex, it is unclear whether a gradient based approach will find the global optimum.

We show that this is indeed the case in the following subsections: In section 2.3, we show that global minima are preserved under the nonconvex reformulation. In section 2.4, show that *nondegenerate* local optima are global optima and that projected gradient descent converges quickly to these. In section 2.5, we show how to modify projected gradient descent to avoid convergence to degenerate local optima and ensure convergence to a global optimum.

## 2.3 Nonconvex reformulation is equivalent to original convex problem

Before arguing that the local minimizers of (3) are equal to the global minimizers of (1), it is important to confirm that the global minimizers are equivalent. Indeed, Theorem 1 confirms this.

**Theorem 1.** *Any feasible solution to* (1) *corresponds to a feasible solution for* (3) *with the same objective value. Furthermore, if* $\mu_i \leq \eta_i$ *for all* $i \in \{1, \ldots, n\}$ *and* $(s, z)$ *feasible to* (3)*, then any feasible solution to* (3) *corresponds to a feasible solution for* (1) *with the same objective value. In which case, the global optimum of* (3) *is same as the global optimum of* (1)*.*

*Proof.* Consider any feasible solution $(s, z)$ to (1). By setting $\theta_i = \frac{z_i - \mu_i}{\eta_i - \mu_i}$ (any $\theta_i \in [0, 1]$ suffices if $\mu_i = \eta_i$) we obtain a feasible solution to (3). On the other hand, if $\mu_i \leq \eta_i$ then (3b) and $\theta_i \in [0, 1]$ implies $\mu_i \leq z_i \leq \eta_i$. □

A sufficient condition for the premise of Theorem 1 to hold is Assumption 3. As Figure 2 shows, if Assumption 3 fails then the nonconvex reformulation can generate infeasible solutions to the original convex optimization problem (1b). Consider the example given by (5) except with $S := [-1.5, 1.5]$ instead of $S := [-1, 1]$. The set of feasible solutions to (1) is enclosed by the two curves. At $s_1 = 1.2$ and $\theta = 1$, $\mu(s_1) > \eta(s_1)$, which is infeasible.

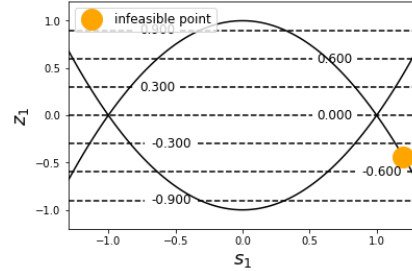

Figure 2: Infeasible: Assumption 3 fails.

## 2.4 Analysis of nondegenerate local optima

This section is devoted to proving that under a nondegeneracy assumption, the first-order stationary points of (3) are global minimizers. Degeneracy issues arise when $\eta_i = \mu_i$. In this situation, if $\theta_i$ changes, then $z$ will remain the same, and therefore from the perspective of the convex formulation, the solution is the same. However, from the perspective of the function $\psi_0$ there is an important difference. In particular, as $\theta_i$ changes the gradient of $\psi_0$ changes. Consequently, certain values of $\theta_i$ may generate spurious local minimizers. Recall example (5), i.e., $S := [-1, 1]$, $f(s_1, z_1) := z_1$, $\eta_1(s_1) := 1 - s_1^2$ and $\mu_1(s_1) := s_1^2 - 1$. In this instance,

$$\psi_0 = \theta_1(1 - s_1^2) + (1 - \theta_1)(s_1^2 - 1) = (1 - 2\theta_1)(s_1^2 - 1), \quad \frac{\partial\psi_0}{\partial s_1} = (1 - 2\theta_1)(2s_1 - 1).$$

As illustrated in Figure 3, the global minimizer is $s_1 = 0, \theta = 0 \Rightarrow z_1 = -1$. If $s_1 \pm 1$ then for all $\theta_1 \in [0, 1]$ we have $z_1 = 0$. Moreover, the points $s_1 \pm 1, \theta_1 \in (0.5, 1]$ are spurious local minimizers.

To see this, note for all $\theta_1 \in [0.5, 1]$, and $s_1 \in S$ that $\psi_0(s_1, \theta_1) \geq 0 = \psi_0(1, \theta_1)$. In contrast, the points $s_1 \pm 1$, $\theta \in [0, 0.5)$ are *not* local minimizers, because for $s_1 \pm 1$ and $\theta_1 \in [0, 0.5)$ we have $\frac{\partial \psi_0}{\partial s_1} > 0$ implying that gradient descent steps move away from the boundary. We conclude that if $\mu_i = \eta_i$ *certain* values of $\theta_i$ could be spurious local minimizers. We emphasize the word *certain* because, as Section 2.5 details, there is always a value of $\theta_i$ that enables escape.

The nondegeneracy assumption we make is that for some $\gamma \geq 0$ the set

$$\mathcal{K}_\gamma(s, \theta) := \left\{ i \in \{1, \ldots, n\} : \quad z = \text{Forward}(s, \theta), \right.$$

$$\left. \eta_i - \mu_i \leq \gamma, \quad \theta_i \left( \frac{\partial \psi_i}{\partial z_i} \right)^+ + (1 - \theta_i) \left( \frac{\partial \psi_i}{\partial z_i} \right)^- > 0 \right\}$$

is empty, where $(\cdot)^+ := \max\{\cdot, 0\}$ and $(\cdot)^- := \min\{\cdot, 0\}$. If the set $\mathcal{K}_0(s, \theta)$ is non-empty then any coordinate $i \in \mathcal{K}_0(s, \theta)$ could be causing a spurious local minimizer. Values of $\gamma$ strictly greater than zero ensures that we do not get arbitrarily close to a degenerate point. We will show this nondegeneracy assumption guarantees that first-order stationary points are global minimizers.

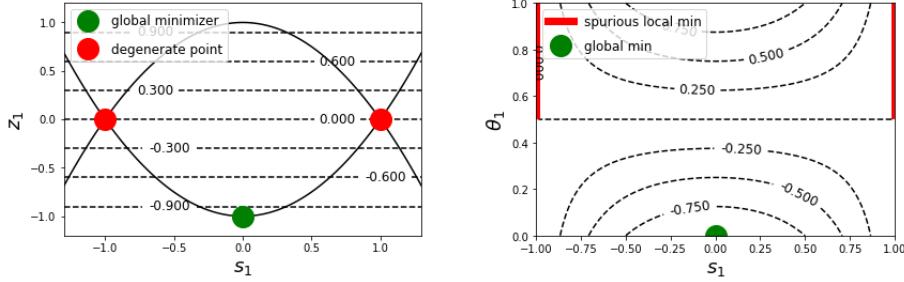

Plot of original convex problem      Plot of nonconvex reformulation $\psi_0(s_1, \theta_1)$

Figure 3: Example of degeneracy causing spurious local minimizers when $s_1 \pm 1$.

While our nondegeneracy assumption holds it will suffice to run PGD which is defined as

$$(s^+, \theta^+) \leftarrow (s, \theta) + \underset{d \in \mathcal{D}(s, \theta)}{\text{argmin}} \, \boldsymbol{\nabla} \psi_0 \cdot d + \frac{L}{2} \|d\|_2^2,$$

where $\mathcal{D}(s, \theta) := \{d : (s, \theta) + d \in S \times [0, 1]^n\}$ is the set of feasible search directions and $L$ is the smoothness of $\psi_0$ (see Definition 1). A useful fact is that PGD satisfies $\psi_0(s^+, \theta^+) \leq \psi_0(s, \theta) - \delta_L(s, \theta)$ for

$$\delta_L(s, \theta) := -\underset{d \in \mathcal{D}(s, \theta)}{\text{minimize}} \, \boldsymbol{\nabla} \psi_0 \cdot d + \frac{L}{2} \|d\|_2^2.$$

See [19, Lemma 2.3.] for a proof. In other words, $\delta_L(s, \theta)$ represents the minimum progress of PGD. Once again for brevity we will denote $\delta_L(s, \theta)$ by $\delta_L$. Note that if $\delta_L$ is zero then we are at a first-order stationary point of $\psi_0$. The remainder of this section focuses on proving that $\delta_L$ provides an upper bound on the optimality gap. To form this bound we use Lagrangian duality. In particular, the Lagrangian of (1) is:

$$\mathcal{L}(s, z, y) := f + \sum_{i=1}^{n} (y_i^+ \mu_i - y_i^- \eta_i - y_i z_i)$$

where $y_i^+ = \max\{y_i, 0\}$, and $y_i^- = \max\{-y_i, 0\}$. We will denote $\mathcal{L}(s, z, y)$ by $\mathcal{L}$. Define,

$$\Delta(s, \theta) := \sum_{i=1}^{n} (y_i z_i - y_i^+ \mu_i + y_i^- \eta_i) + \sup_{(\hat{s}, \hat{z}) \in S \times Z} \boldsymbol{\nabla}_{s,z} \mathcal{L} \cdot (\hat{s} - s, \hat{z} - z) \qquad (7)$$

with $z = \text{FORWARD}(s, \theta)$ and $y_i = \frac{\partial \psi_i}{\partial z_i}$. If $(s, z)$ is feasible to (1) we conclude $\Delta(s, \theta)$ is a valid duality gap, i.e., provides global guarantees, because by duality, convexity and (7),

$$f_* \geq \inf_{(\hat{s}, \hat{z}) \in S \times Z} \mathcal{L}(\hat{s}, \hat{z}, y) \geq \mathcal{L} + \inf_{(\hat{s}, \hat{z}) \in S \times Z} \boldsymbol{\nabla}_{s,z} \mathcal{L} \cdot (\hat{s} - s, \hat{z} - z) = f - \Delta(s, \theta). \qquad (8)$$

To compute $\Delta(s,\theta)$, one needs to be able to efficiently minimize a linear function over the set $Z$. For this reason, one should choose $Z$ to have a simple form (i.e., bound constraints).

**Assumption 4.** *There exists a constant $c > 0$ such that $\|\eta - \mu\|_2 + D_s\|\nabla_s\eta - \nabla_s\mu\|_2 + D_z\|\nabla_z\eta - \nabla_z\mu\|_2 \le c$ for all $(s,z)$ that are feasible to* (1b).

In Assumption 4, observe that $\nabla_s\eta - \nabla_s\mu$ and $\nabla_z\eta - \nabla_z\mu$ are matrices so $\|\cdot\|_2$ is the spectral norm. Also, note that Assumption 1 and 2 imply that Assumption 4 must hold. However, we add Assumption 4 because it makes the constant $c$ explicit.

**Lemma 1** (Nondegenerate first-order stationary points are optimal)**.** *Suppose Assumption 2 and 4 hold. Suppose also that $\mathcal{K}_\gamma(s,\theta) = \emptyset$ with $\gamma \in (0,\infty)$, and that $\delta_L \le L/2$. Then $\Delta(s,\theta)^2 \le L\left(D_s\sqrt{2} + 2\gamma^{-1}c\right)^2 \delta_L$.*

In the nondegenerate case (i.e., $\mathcal{K}_\gamma(s,\theta) = \emptyset$), $\delta_L$ upper bounds $\Delta(s,\theta)$. In particular, as Lemma 1 demonstrates small progress by gradient steps implies small duality gaps. The proof of Lemma 1 appears in Section C.1 and is technical. The core part of the proof of Lemma 1 is bounding $\theta_i y_i^+ + (1-\theta_i)y_i^-$ for $y_i = \frac{\partial\psi_i}{\partial z_i}$ in terms of $\gamma^{-1}$ and $\delta_L$. When $\theta_i y_i^+ + (1-\theta_i)y_i^- \approx 0$ one can show that $\Delta(s,\theta) \approx \sup_{\hat{s}\in S} \nabla_s\mathcal{L}\cdot(\hat{s}-s) \approx \sup_{\hat{s}\in S}\nabla_s\psi\cdot(\hat{s}-s) \le D_s\sqrt{2L\delta_L}$.

### 2.4.1 Analysis of projected gradient descent

Lemma 1 provides the tool we need to prove the convergence of PGD in the nondegenerate case. The algorithm we analyze (Algorithm 1) includes termination checks for optimality. Furthermore, the PGD steps can be replaced by any algorithm that makes at least as much function value reduction as PGD would make in the worst-case. For example, gradient descent with a backtracking line search and an Armijo rule [20, Chapter 3], or a safeguarded accelerated scheme [21] would suffice.

---

**Algorithm 1** Local search algorithm for minimizing $\psi_0$ in the nondegenerate case.

---

1: **function** SIMPLE-PSI-MINIMIZATION$(s^1,\theta^1,\epsilon)$
2:     Suppose $\psi_0$ is $L$-smooth. Note $L \in (0,\infty)$ need not be known.
3:     **for** $k = 1,\ldots,\infty$ **do**
4:         *Termination checks:*
5:         **if** $\Delta(s^k,\theta^k) \le \epsilon$ **then**
6:             *Found an $\epsilon$-optimal solution:*
7:             **return** $(s^k,\theta^k)$
8:         **end if**
9:         *Reduce the function at least as much as PGD would:*
10:        $(s^{k+1},\theta^{k+1}) \in \{(s,\theta): \psi_0(s,\theta) \le \psi_0(s^k,\theta^k) - \delta_L(s^k,\theta^k)\}$
11:     **end for**
12: **end function**

---

**Theorem 2** (PGD converges to global minimizer under nondegeneracy assumption)**.** *Suppose Assumption 2, 3 and 4 hold. Suppose $\psi_0$ is $L$-smooth, $\epsilon,\gamma \in (0,\infty)$, $(s^1,\theta^1) \in S\times[0,1]^n$, and $\mathcal{K}_\gamma(s^k,\theta^k) = \emptyset$ for all iterates of the algorithm* SIMPLE-PSI-MINIMIZATION$(s^1,\theta^1,\epsilon)$. *Then, the algorithm terminates after at most*

$$1 + \frac{2\Delta(s^1,\theta^1)}{L} + \frac{L\left(D_s\sqrt{2} + 2c\gamma^{-1}\right)^2}{\epsilon} \quad \text{iterations.}$$

See Section C.2 for a proof of Theorem 2. The proof of Theorem 2 directly utilizes Lemma 1 using standard techniques, almost identical to the proof of convergence for gradient descent in the convex setting [13, Theorem 2.1.13].

**Remark 2.** *It is worth discussing the premise in Theorem 2 that $\psi_0$ is $L$-smooth. The composition of smooth functions is smooth, implying $\psi_0$ is smooth. Moreover, since $S\times[0,1]^n$ is a bounded set we deduce that $\psi_0$ is $L$-smooth for some $L > 0$. Therefore the premise that $\psi_0$ is $L$-smooth is valid. However, the value of $L$ could be extremely large, for example, if $\eta_i(s,z_{1:i-1}) = \mu_i(s,z_{1:i-1}) = 2z_{i-1}$ for $i > 1$, $\eta_1(s) = \mu_1(s) = s_1$, and $f(s,z) = \frac{1}{2}z_n^2$ then $\psi_0(s,\theta) = \frac{1}{2}(2^n s)^2$ and $L = 4^n$. Note this occurs despite the fact that each component function is well-behaved (i.e., $\eta_i,\mu_i,f$ are 1-smooth and 2-Lipschitz with respect to the Euclidean norm).*

**Remark 3.** *Consider* (2), *the hard example for standard first-order methods. Note that starting from the origin (i.e., $x_1 = 0$, $\theta = \mathbf{0}$), then for sufficiently large step size PGD on $\psi_0$ will take exactly one iteration to find the optimal solution ($x_1 = 1, \theta = \mathbf{1}$).*

**Remark 4.** *Suppose that we are solving a neural network verification problem (Section 3 and A.2). Then this approach is strongly related to adversarial attack heuristics. In particular, freezing $\theta = \mathbf{0}$ in* SIMPLE-PSI-MINIMIZATION *yields a typical gradient based attack on the network [22].*

### 2.5 Analysis of degenerate local optima

Section 2.4 proved convergence of PGD to the global minimizer under a nondegeneracy assumption (i.e., $\mathcal{K}_\gamma(s^k, \theta^k) = \emptyset$). This section develops a variant of PGD that requires no degeneracy assumptions but still converges to the global minimizer.

#### 2.5.1 Escaping exact local minimizers

Our main result, presented in Section 2.5.2, proves convergence under minimal assumptions. The key to the result is developing an algorithm for escaping basins of local minimizers. However, the algorithm and analysis is very technical. To give intuition for it this section considers the easier case of escaping *exact* local minimizers (Lemma 2).

The high level idea is illustrated in Figure 4b. Recall from Figure 3 that if we are at a spurious local minimizer then the set $\mathcal{K}_\gamma(s, \theta)$ must be nonempty. In particular, in this instance the set $\mathcal{K}_0(s, \theta) = \{1\}$ is nonempty. In this setting, $\theta_1$ corresponds to an edge that we can move along where $\psi_0(s, \theta)$ is constant. ESCAPE-EXACT-LOCAL-MIN$(s, \theta)$ moves us along this edge from $(s, \theta)$ to $(s, \hat{\theta})$ at which $\mathcal{K}_0(s, \hat{\theta})$ is empty and therefore we have escaped the local minimizer.

1: **function** ESCAPE-EXACT-LOCAL-MIN$(s, \theta)$
2:     $z = $ FORWARD$(s, \theta)$, $\hat{\theta} \leftarrow$ copy$(\theta)$
3:     **for** $i = n, \ldots, 1$ **do**
4:         **if** $i \in \mathcal{K}_0(s, \theta_{1:i}, \hat{\theta}_{i+1:n})$ **then**
5:         $\hat{\theta}_i = \begin{cases} 0 & \frac{\partial \psi_i(s, z_{1:i}, \hat{\theta}_{i+1:n})}{\partial z_i} > 0 \\ 1 & \frac{\partial \psi_i(s, z_{1:i}, \hat{\theta}_{i+1:n})}{\partial z_i} < 0 \end{cases}$
6:         **end if**
7:     **end for**
8:     **return** $(s, \hat{\theta})$
9: **end function**

(a) Algorithm

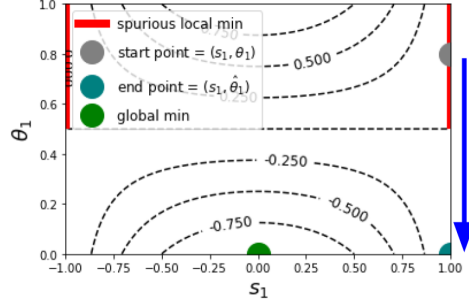

(b) The high level idea of the algorithm is shown by the blue arrow.

Figure 4: An algorithm for escaping exact local minimizers

**Lemma 2** (Escaping exact local minimizers). *Suppose that Assumption 1 holds and let $(s, \hat{\theta}) = $ ESCAPE-EXACT-LOCAL-MIN$(s, \theta)$. Then* FORWARD$(s, \theta) = $ FORWARD$(s, \hat{\theta})$, *and* $\mathcal{K}_0(s, \hat{\theta}) = \emptyset$.

*Proof.* By the definition of $\mathcal{K}_0$, if $i \in \mathcal{K}_0(s, \theta)$ then $\eta_i = \mu_i$. Therefore FORWARD$(s, \theta_{1:i-1}, \hat{\theta}_{i:n}) = $ FORWARD$(s, \theta_{1:i}, \hat{\theta}_{i+1:n})$, and by induction FORWARD$(s, \theta) = $ FORWARD$(s, \hat{\theta})$.

Next, we show that $i \notin \mathcal{K}_0(s, \theta_{1:i-1}, \hat{\theta}_{i:n})$. If $i \notin \mathcal{K}_0(s, \theta_{1:i}, \hat{\theta}_{i+1:n})$ then $\theta_i = \hat{\theta}_i$ so the result trivially holds. On the other hand, if $i \in \mathcal{K}_0(s, \theta_{1:i}, \hat{\theta}_{i+1:n})$ then by definition of $\hat{\theta}_i$,

$$\hat{\theta}_i \left( \frac{\partial \psi_i(s, z_{1:i}, \hat{\theta}_{i+1:n})}{\partial z_i} \right)^+ + (1 - \hat{\theta}_i) \left( \frac{\partial \psi_i(s, z_{1:i}, \hat{\theta}_{i+1:n})}{\partial z_i} \right)^- = 0 \qquad (9)$$

which implies $i \notin \mathcal{K}_0(s, \theta_{1:i-1}, \hat{\theta}_{i:n})$. Further note that FORWARD$(s, \theta_{1:i-1}, \hat{\theta}_{i:n}) = $ FORWARD$(s, \theta_{1:i}, \hat{\theta}_{i+1:n})$ implies if $j \leq i$ and $j \notin \mathcal{K}_0(s, \theta_{1:j-1}, \hat{\theta}_{j:n})$ then $j \notin \mathcal{K}_0(s, \theta_{1:i-1}, \hat{\theta}_{i:n})$. By induction we deduce $\mathcal{K}_0(s, \theta_{1:i-1}, \hat{\theta}_{i:n}) \subseteq \{1, \ldots, i-1\}$ and hence $\mathcal{K}_0(s, \hat{\theta})$ is empty. $\square$

A critical feature of ESCAPE-EXACT-LOCAL-MIN$(s, \theta)$ is that we work backwards (i.e., $i = n, \ldots, 1$ rather than $i = 1, \ldots, n$). This is critical because if we work forwards instead of backwards then (9)

would become

$$\hat{\theta}_i \left( \frac{\partial \psi_i(s, z_{1:i}, \theta_{i+1:n})}{\partial z_i} \right)^+ + (1 - \hat{\theta}_i) \left( \frac{\partial \psi_i(s, z_{1:i}, \theta_{i+1:n})}{\partial z_i} \right)^- = 0$$

which, due to the replacement of $\theta$ with $\hat{\theta}$ inside $\psi_i$, is insufficient to establish $\mathcal{K}_0(s, \hat{\theta})$ is empty.

Finally, we remark that $g_i := \frac{\partial \psi_i(s, z_{1:i}, \hat{\theta}_{i+1:n})}{\partial z_i}$ can be computed via the recursion

$$g_i \leftarrow \frac{\partial f}{\partial z_i} + \sum_{j=i+1}^n g_j \left( \hat{\theta}_j \frac{\partial \eta_j}{\partial z_i} + (1 - \hat{\theta}_j) \frac{\partial \mu_j}{\partial z_i} \right),$$

and therefore calling ESCAPE-EXACT-LOCAL-MIN takes the same time as computing $\boldsymbol{\nabla}_\theta \psi_0$.

### 2.5.2 Escaping the basin of a local minimizer

If we modify SIMPLE-PSI-MINIMIZATION to run ESCAPE-EXACT-LOCAL-MIN$(s, \theta)$ whenever the set $\mathcal{K}_0(s^k, \theta^k)$ is nonempty then we would escape exact local minimizers. However, that does not exclude the possibility of asymptotically converging to a local minimizer. Therefore we need a method that will escape the basin of a local minimizer. In particular, we must be able to change the value of the $\theta_i$ variables with $i \in \mathcal{K}_\gamma(s, \theta)$ for $\gamma > 0$. This, however, introduces technical complications because if $\eta_i > \mu_i$ then as we change $\theta_i$ the value of $z_{i:n}$ could change.

Due to these technical complications we defer the algorithm and analysis to Appendix D, and informally state the main result here. The proof of Theorem 3 appears in Appendix D.1. The discussion given in Remark 2 also applies to Theorem 3 and means that the constant $C$ could be large.

**Theorem 3.** *Suppose that Assumptions 1, 2, and 3 hold. Then there exists an algorithm obtaining an $\epsilon$-duality gap after $C\epsilon^{-3} + 1$ computations of $\boldsymbol{\nabla}\psi_0$ where $C$ is a problem dependent constant.*

## 3 Experiments

We evaluate our method on robustness verification of models trained on CIFAR10 [23]. We benchmark on three sizes of networks trained with adversarial training [24]. The tiny network has two fully connected layers, with 100 units in the hidden layer. The small network has two convolutional layers, and two fully connected layers, with a total of 8308 hidden units. The medium network has four convolutional layers followed by three fully connected layers, with a total of 46912 hidden units.

Verification of these networks is relaxed to a stage-wise convex problem (Appendix A.2). We compare three strategies for solving this relaxation: (i) NonConvex, our nonconvex reformulation using SIMPLE-PSI-MINIMIZATION augmented with momentum and backtracking linesearch, (ii) DeepVerify [6] (DV) that performs Lagrangian relaxation on the bound computation problem, (iii) a direct encoding of the relaxation into CVXPY [28], with SCS [26] and ECOS [27] backends[2]. We

| ReLU Activation | Average Bound | | | Runtime (ms) | | |
|---|---|---|---|---|---|---|
| | Tiny | Small | Medium | Tiny | Small | Medium |
| IBP [25] | 17.0 | 743 | 2.4e+6 | 5.5 | 3.1 | 3.3 |
| DeepVerify [6] | 13.7 | 544 | 1.6e+6 | 349 | 711 | 1.1e+3 |
| NonConvex (Ours) | 5.68 | 434.9 | 1.5e+6 | 91.2 | 177 | 175 |
| CVXPY (SCS) [26] | 5.64 | - | - | 1.7e+5 | - | - |
| CVXPY (ECOS) [27] | 5.64 | - | - | 4.3e+4 | - | - |

| SoftPlus Activation | Average Bound | | | Runtime (ms) | | |
|---|---|---|---|---|---|---|
| | Tiny | Small | Medium | Tiny | Small | Medium |
| IBP [25] | 18.3 | 6.5e+3 | 2.0e+9 | 4 | 2.5 | 3.3 |
| DeepVerify [6] | 13.7 | 5.1e+3 | 1.5e+9 | 414 | 855 | 1.7e+3 |
| NonConvex (Ours) | 5.97 | 3.93e+3 | 1.3e+9 | 7.8 | 65 | 214 |
| CVXPY (SCS) [26] | 5.97 | - | - | 2.9e+5 | - | - |

Table 1: **Benchmark** For each model, we report the average bound achieved on the adversarial objective and the average runtime in milliseconds to obtain it, over the CIFAR-10 test set. IBP [25] does not perform any optimization so it has an extremely small runtime but the bounds it generates are much weaker. The off-the-shelf solvers are significantly slower than the first-order methods DeepVerify and NonConvex and were not feasible to run beyond the tiny network.

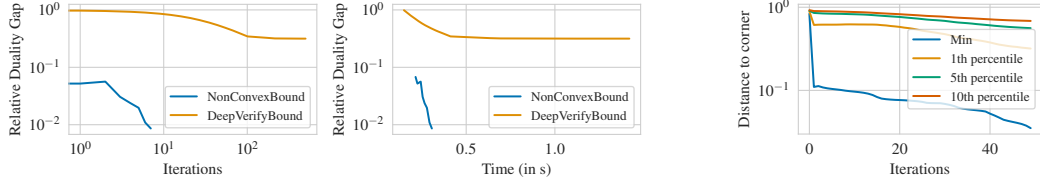

(a) Evolution of the relative duality gap as a function of time or number of iteration, for the NonConvex and DeepVerify Solver.

(b) Distribution of distance to potentially degenerate points.

Figure 5: Evaluation on the Medium-sized network with SoftPlus activation function

terminate (i) after 50 iteration or when the relative duality gap is less than $10^{-2}$, (ii) after 500 iterations or when its dual value is larger than the final value of NonConvex (NC) (details in Appendix F).

Table 1 shows that, compared with the specialized first-order method DV, our method is faster by a factor between 3 and 50 depending on the network architecture, and always produces tighter bounds. As the two methods solve problems that have the same optimal value, we hypothesize that the discrepancy is because the Lagrangian relaxation of DV contains an inner-maximization problem that makes its objective extremely non-smooth, slowing convergence.

In most problems, DV reaches the imposed iterations limit before convergence. This is quantified in Table 2 where we show that beyond the tiny network, DV does not reach a small enough dual gap to achieve early stopping. On the other hand, we observe that for NC, the scale of the network does not significantly impact the required number of iterations. Figure 5a shows an example of the evolution of the computed bound, where we can see that the objective of DV plateaus, while NC converges in few iterations. Since the time per iteration for both methods is roughly the same, our runtime is lower.

After a single iteration, the duality gap achieved by our method is considerably smaller. The variables of DV exist on an unbounded feasible domain and appropriate initial values are therefore difficult to estimate, leading to large initial duality gap. Our method does not suffer from this problem, as all our variables are constrained between 0 and 1, and we can therefore initialize them all to 0.5, which empirically gives us good performance.

**Nondegeneracy in practice.** In Section 2.4, we described a simple version of our algorithm under the assumption that the algorithm does not enter a degenerate region. In the context of Neural Network verification, due to the structure of the problem, the only possibility for a small gap between $\eta_i - \mu_i$ is at the boundary of the feasible domain of the convex hull relaxation of activation. Even points close to the corner are not necessarily degenerate as they also need to satisfy a condition on the gradients. Throughout optimization, we measure $\frac{\min\{z_i - l_i, u_i - z_i\}}{u_i - l_i}$ where $l_i$ and $u_i$ are lower and upper bounds on

|  | Early stopping % | | Avg iteration count | |
|---|---|---|---|---|
|  | DV | NC | DV | NC |
| Tiny ReLU | 37% | 73% | 384 | 18 |
| Small ReLU | 0% | 97% | 500 | 9 |
| Medium ReLU | 63% | 100% | 284 | 5 |
| Tiny SoftPlus | 14 % | 100% | 467 | 4 |
| Small SoftPlus | 0 % | 100% | 500 | 7 |
| Medium SoftPlus | 0 % | 59% | 500 | 25 |

Table 2: Proportion of bound computations on CIFAR-10 where the algorithm converges within the iteration budget, and average number of iterations.

$z_i$ (corresponding to the corners), as shown in Figure 5b. We can observe that this value is strictly positive for all $i$ which means we are not entering the degenerate region. This explains why, for these problems, SIMPLE-PSI-MINIMIZATION was able to converge to good solutions.

**Conclusion:** We have developed a novel algorithm for a class of stage-wise convex optimization problems. Our experiments showed that our algorithm is efficient at solving standard relaxations of neural network verification problems. We believe that these results will generalize to stronger relaxations [29], as well as other stage-wise convex problems such as those arising in optimal control and generalized isotonic regression.

## Broader Impact

Our work leads to new scalable algorithms for verifying properties of neural networks and solve certain kinds of structured regression problems. On the positive side, these can have an impact in terms of better methods to evaluate the reliability and trustworthiness of state of the art deep learning systems, thereby catching any unseen failure modes and preventing undesirable consequences of deep learning models. On the negative sign, the algorithms are agnostic to the type of properties being verified and may facilitate abuses by allowing attackers to verify that their attacks can reliably induces specific failure modes in a deep learning model. Further, any applications of these techniques is reliant on carefully designing desirable specifications or properties of a deep learning model - if this is not done carefully, even systems that are verifiable with these algorithms may exhibit undesirable behavior (arising from bias in the data or the specification).

## Acknowledgments and Disclosure of Funding

We thank Miles Lubin for establishing the connections between the authors and helpful feedback on the paper. We'd also like to thank Ross Anderson, Christian Tjandraatmadja, and Juan Pablo Vielma for helpful discussions.

## Footnotes

[2]We also ran tests on an internal primal-dual hybrid gradient implementation. It was not remotely competitive (failing to converge after 100,000 iterations on trialed instances) so we did not include it in the results.

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
