[Supplementary Material]

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

# A  Examples of optimization problems with this structure

## A.1  Linear quadratic control and extensions

Linear quadratic control problems [1] take the form:

$$\underset{x}{\text{minimize}} \frac{1}{2} \sum_{t=1}^{M} x(t)^T Q(t) x(t) + u(t)^T R(t) u(t) \tag{10a}$$

$$x(t+1) = A(t)x(t) + B(t)u(t) \tag{10b}$$

$$x(0) = x_{\text{initial}} \tag{10c}$$

where $A(t), B(t) \in \mathbf{R}^{n \times n}$ are matrices, $Q(t), R(t) \in \mathbf{R}^{n \times n}$ are symmetric positive definite matrices, and $x(t) \in \mathbf{R}^n$ represents the system state, $u(t) \in \mathbf{R}^n$ the input, the initial system state is $x_{\text{initial}} \in \mathbf{R}^n$, and the positive integer $M$ is the number of time steps. This problem can be solved by dynamic programming using $O(Mn^3)$.

Another approach [30] is to reformulate by eliminating the $x$ variables by forward propagation, thereby rewriting the problem as

$$\underset{u}{\text{minimize}} \, h(u). \tag{11}$$

The gradient of (11) can be computed by backpropagation which takes time proportional to the total number of non-zeros in $A(t)$, $B(t)$, $Q(t)$, and $R(t)$. One can therefore solve this problem using gradient descent and due to the lower iteration cost, potentially find an approximate minimizer faster than using dynamic programming. The function $h(u)$ is a convex quadratic, which implies gradient descent finds the global minimizer. It is worth noting that applying our nonconvex reformulation to (10) by letting $s \leftarrow u$ and $z \leftarrow x$ yields $h(u) = \psi_0(u, \theta)$. Therefore, for linear quadratic control our approach and the approach of Lasdon et al. [30] are essentially equivalent.

However, one benefit of our approach is that we can tackle a wider range of problems than these classical methods. For example, we can support more complex dynamics where $x(t+1)$ is wedged between a convex and concave function.

## A.2  Verification of neural networks robustness to adversarial attacks

To provide guarantees on the behaviour of neural networks, there has been a surge of interest in verifying that the output classification of a trained model remains stable when the input is slightly perturbed (adversarially) [5–7]. In particular, consider an input $s_0$ with label $c^*$. We wish to show there exists no adversarial example close to $s_0$ such that the network outputs $c \neq c^*$. Define $S$ as the restriction of the input domain over which we want to perform verification. In the context of robustness to adversarial attacks, this would typically correspond to $S = \{s \mid \|s - s_0\|_\infty \leq \epsilon\}$. The set of feasible activation values for a feedfoward neural network for any input $s \in S$ satisfy,

$$s \in S \tag{12a}$$

$$z_{1:j(1)} = \sigma(W_0 s + b_0) \tag{12b}$$

$$z_{j(k)+1:j(k+1)} = \sigma(W_k z_{j(k-1)+1:j(k)} + b_k), \quad \forall k \in \{1, \ldots, K-1\}, \tag{12c}$$

where $k$ represents layers, $j(0), \ldots, j(K)$ partitions the vector $z$ such that $z_{j(k-1)+1:j(k)}$ is the activation values for layer $k$, $W_k$ is the weight matrix for the $k$th layer, and $\sigma$ represents the activation function (e.g., ReLU). By splitting the matrices $W_k$ into a sequence of vectors $w_i$ and the vectors $b_k$ into a sequence of numbers $h_i$ this can be rewritten in the form

$$s \in S \tag{13a}$$

$$z_i = \sigma([s, z_{1:i-1}] \cdot w_i + h_i) \quad \forall i \in \{1, \ldots, n\}. \tag{13b}$$

To make the definitions precise, $w_i = [\mathbf{0}, [W(k)]_{i-j(k-1)}]$ where $k$ is the unique solution to $j(k-1) + 1 \leq i \leq j(k)$, and $[W(k)]_{i-j(k-1)}$ denotes the $i - j(k-1)$th row of $W(k)$; $h_i = [b_k]_{i-j(k-1)}$. Note that (13) is more general than (12) as it could capture more than just feedforward networks.

We now describe a procedure to verify the network. Let $C$ be the set of possible output classes from the network and $v_c$ a weight vector for each class $c \in C$. Typically neural networks classify an example according to the rule

$$\underset{c \in C}{\text{argmax}} \, v_c \cdot z.$$

Usually, $v_c$ is a sparse vector with zeros in all entries except those corresponding to the last layer of the network.

Therefore to verify that the network will output class $c^*$ for all inputs in $S$ it suffices to solve

$$\underset{z}{\text{minimize}} \ (v_{c^*} - v_c) \cdot z \quad \text{subject to} \quad (13),$$

for each $c \in C \setminus \{c^*\}$. If the minimum value of each of these subproblems is positive then the network is robust to adversarial perturbations.

Unfortunately, this problem is intractable as the feasible region given by (13) is nonconvex, and moreover the problem is in general NP-hard [31]. However, this does not preclude the possibility of verifying the neural network by forming a convex relaxation of (13). To form this convex relaxation of (13), we need lower and upper bounds on the possible values for each value of $[s, z_{1:i-1}] \cdot w_i$. These bounds can be obtained either by optimization over the partially constructed problem or by simple bound propagation [25]. Let us denote these bounds by $l_i$ and $u_i$. In the case where $\sigma$ is a ReLU we define the convex relaxation in the form of (1) with

$$f(s, z) = (v_c - v_{c'}) \cdot z$$

and for all $i \in \{1, \ldots, n\}$,

$$\mu_i(s, z_{1:i-1}) = \sigma([s, z_{1:i-1}] \cdot w_i + h_i) = \max\{[s, z_{1:i-1}] \cdot w_i + h_i, 0\}$$

$$\eta_i(s, z_{1:i-1}) = \begin{cases} \frac{u_i}{u_i - l_i}([s, z_{1:i-1}] \cdot w_i + h_i - l_i) & 0 \in [l_i, u_i] \\ [s, z_{1:i-1}] \cdot w_i + h_i & l_i \geq 0 \\ 0 & u_i \leq 0 \end{cases}$$

where $\mu_i$ and $\eta_i$ are depicted in Figure 6. Since ReLU is a convex function, this feasible region is convex. Definition of the constraints corresponding to the convex hull relaxation of different type of non-linearities have been previously published in the literature [8, 32].

Due to the way that the lower and upper bounds are constructed, this convex relaxation satisfies Assumption 3. In particular, if we form the bounds by optimizing over the partially constructed problem, i.e.,

$$l_j = h_j + \underset{(s,z) \in S \times \mathbf{R}^n}{\text{minimize}} \ [s, z_{1:j-1}] \cdot w_j \quad \text{s.t.} \quad \mu_i(s, z_{1:i-1}) \leq z_i \leq \eta_i(s, z_{1:i-1}) \quad \forall i \in \{1, \ldots, j-1\}$$

$$u_j = h_j + \underset{(s,z) \in S \times \mathbf{R}^n}{\text{maximize}} \ [s, z_{1:j-1}] \cdot w_j \quad \text{s.t.} \quad \mu_i(s, z_{1:i-1}) \leq z_i \leq \eta_i(s, z_{1:i-1}) \quad \forall i \in \{1, \ldots, j-1\}$$

then we can see that given we have a feasible solution $[s, z_{1:j-1}]$ to $\mu_i(s, z_{1:i-1}) \leq z_i \leq \eta_i(s, z_{1:i-1}) \ \forall i \in \{1, \ldots, j-1\}$ then $[s, z_{1:j-1}] \cdot w_j + h_j \in [l_i, u_i]$ which implies that for $0 \notin [l_i, u_i]$ that $\eta_j(s, z_{1:j-1}) - \mu_j(s, z_{1:j-1}) \geq 0$ by definition and if $0 \in [l_i, u_i]$ then

$$\eta_j(s, z_{1:j-1}) - \mu_j(s, z_{1:j-1}) = \frac{u_i}{u_i - l_i}([s, z_{1:i-1}] \cdot w_i - l_i) - \max\{[s, z_{1:i-1}] \cdot w_i, 0\}$$

$$\geq \min\left\{\frac{u_i}{u_i - l_i}(u_i - l_i), \frac{u_i}{u_i - l_i}(l_i - l_i)\right\}$$

$$= 0$$

as required to establish Assumption 3 (intuition for this can be given by contrasting Figure 2 with Figure 6). By a similar argument, simple bound propagation [25] to compute $l_i$ and $u_i$ will also guarantee Assumption 3 holds. In general, if a relaxation is constructed in an inductive fashion Assumption 3 tends to naturally hold.

There exists tighter linear programming bounds for neural network verification that require an (implicit) exponential number of inequalities [29]. Ignoring smoothness issues, these also can be cast into our framework. Using our approach to solve these linear programs is an interesting avenue for future work. These types of relaxations also have applications in reinforcement learning [33].

**Related work on Deep Network verification** The convex relaxation was proposed by Ehlers [Ehlers, ATVA 2017] but the high computational cost of solving it with off-the-shelf LP solvers was prohibitive for large instances. A large number of papers such as IBP [34], DeepPoly [35], Crown [36], Neurify [37], LP-relaxed-dual [7] focused on looser relaxations to allow for fast, closed

Figure 6: Comparison of original feasible region with convex relaxation when $\sigma$ is ReLU.

form solutions of the bound computation problem scaling to larger networks. In parallel, work was done to reformulate the optimization problem to allow the use of better algorithms: DeepVerify [38] introduced an unconstrained dual reformulation of the non-convex problem and showed equivalence with the convex relaxation. Proximal [32] performed lagrangian decomposition and used proximal methods to solve the problem faster. The application of our method to network certification follows this research direction of speeding up the computation of network bounds without compromising on tightness.

### A.3 Finite horizon Markov decision processes with cost function controlled by an adversary

Consider a finite horizon Markov Decision Process with uncertain rewards. In particular, at each time stage $t \in \{1, \dots, T\}$ we take an action $a \in A$ and move from state $k \in K$ to a state $k' \in K$ with probability $P_{a,k,k',t}$ and earn reward $R_{a,k,t}$. Further suppose the rewards are uncertain but we know $R \in \mathcal{R}$ where $\mathcal{R}$ is a convex set. We wish to know what the optimal policy is given we start in state 1. This can be written as the following optimization problem

$$\min_{R \in \mathcal{R}, v} \quad v_{1,1} \tag{14a}$$

$$\max_{a \in A} \sum_{k' \in K} P_{a,k,k',t} v_{k',t+1} + R_{a,k,t} \leq v_{k,t} \tag{14b}$$

where $v_{k,T} = 0$. McMahan et al. [2] develops a specialized algorithm for this problem. This algorithm is used for robots playing laser tag [39].

We can also cast this problem in our framework. Suppose we have an upper bound on the rewards $R_{a,k,t}^{\max}$ ($R_{a,k,t}^{\max} \geq R_{a,k,t}$ for all $R \in \mathcal{R}$) then applying standard dynamic programming we can compute $v_{k,t}^{\max}$ allowing us to rewrite (14) in the form,

$$\min_{R \in \mathcal{R}, v} \quad v_{1,1}$$

$$\max_{a \in A} \sum_{k' \in K} P_{a,k,k',t} v_{k',t+1} + R_{a,k,t} \leq v_{k,t} \leq v_{k,t}^{\max}.$$

For this problem one can show Assumption 2 and 3 are satisfied. The only issue is that $\max_{a \in A} \sum_{k' \in S} P_{a,k,k',t} v_{k',t+1} + R_{a,k,t}$ is a nonsmooth function. Although, as we mention in Remark 1, this problem is likely surmountable.

### A.4 Generalized Isotonic regression

Classic isotonic regression considers the following problem

$$\min_z \sum_{i=1}^{n} (z_i - y_i)^2 \quad \text{s.t.} \quad z_j \leq z_i \quad \forall j \prec i. \tag{15}$$

This problem has applications in genetics [3, 4], psychology [40], and biology [41]. When $\prec$ is a total ordering then there exists efficient algorithms that use only linear time [42]. However, for the general case developing efficient algorithms is an area of active research [4].

Our approach offers a new way of solving these problems. In particular, if we add the mild assumption that $z_i$ is bounded in $[l, u]$ then (15) reduces to

$$\min_z \sum_{i=1}^{n} (z_i - y_i)^2 \quad \text{s.t.} \quad \max\{z_j, l\} \leq z_i \leq u \quad \forall j \prec i. \tag{16}$$

Note that if $\prec$ is a partial ordering then there exists a total ordering that is consistent with this partial ordering. We can represent such a total ordering by a permutation $\pi : \{1, \ldots, n\} \to \{1, \ldots, n\}$ where $i \preceq j \Rightarrow \pi(i) \leq \pi(j)$. Therefore, without loss of generality assume that that $i \preceq j \Rightarrow i \leq j$. This allows us to write the problem in the form of (1), as

$$\min_z \sum_{i=1}^n (z_i - y_i)^2 \quad \text{s.t.} \quad \mu_i(z_{1:i-1}) \leq z_i \leq \eta_i(z_{1:i-1}), \quad \forall i \in \{1, \ldots, n\},$$

where $\mu_i(z_{1:i-1}) = \max\left\{l, \underset{j:j\prec i}{\text{maximum}} \, z_j\right\}$ and $\eta_i(z_{1:i-1}) = u$. Given $\mu_i(z_{1:i-1}) \leq \eta_i(z_{1:i-1})$ for all $i < j$ then $z_i \leq u$ which implies that $\eta_j(z_{1:j-1}) \leq u$, i.e., Assumption 3 holds.

# B  Sketch of lower bounds

Here we briefly sketch how to use to show that solving (2) requires at least $n-1$ iterations of saddle point methods. We only provide a sketch since very similar results are already known [43, 44]. Before reading this section we recommend reading a standard reference on lower bounds (e.g., [45, Section 3.5], [13, Section 2.1.2]) and a standard reference on saddle point methods (e.g., [45, Section 5.2]).

We can reformulate (2) as a saddle point problem as follows

$$\underset{x \in X}{\text{minimize}} \quad \underset{y \in Y}{\text{maximize}} - e_n^T x + y^T A x$$

where $e_n$ is a vector with a one in the $n$th entry and all other values zero, $X := \{x \in \mathbf{R}^n : x \geq -1, x_1 \in [0, 1]\}$, $Y := \{y \in \mathbf{R}^{n-1} : y \geq 0\}$ and

$$A = \begin{pmatrix} 1 & -1 & & \\ & 1 & -1 & \\ & & \cdots & \\ & & & 1 & -1 \end{pmatrix}.$$

After appropriate reindexing the iterates $\{(x^t, y^t)\}_{t=0}^\infty$, saddle point algorithms such as primal-dual hybrid gradient [10] and mirror-prox [9] satisfy

$$\hat{x}_{t+1} \in x_t + \text{span}(A^T y_0, \ldots, A^T y_t)$$
$$x_{t+1} = \Pi_X(\hat{x}_{t+1})$$
$$\hat{y}_{t+1} \in y_t + \text{span}(A x_0, \ldots, A x_t)$$
$$y_{t+1} = \Pi_Y(\hat{y}_{t+1})$$

where $\Pi_X$ and $\Pi_Y$ projects onto the set $X$ and $Y$ respectively.

Define $Z_X^t = \{x \in \mathbf{R}^n : x_i = 0, \forall i \in \{1, \ldots, n-t\}\}$ and $Z_Y^t = \{y \in \mathbf{R}^{n-1} : y_i = 0, \forall i \in \{1, \ldots, n-t-1\}\}$. Now, if $t \in \{0, \ldots n-2\}$, $x^0, \ldots, x^t \in Z_X^t$ and $y^0, \ldots, y^t \in Z_Y^t$ then

- $\hat{x}^{t+1} \in x^0 + \text{span}(A^T y^0, \ldots, A^T y^t) \subseteq x^t + Z_X^{t+1} = Z_X^{t+1} \Rightarrow x^{t+1} \in Z_X^{t+1}$.
- $\hat{y}^{t+1} \in y^0 + \text{span}(A x^0, \ldots, A x^t) \subseteq y^t + Z_Y^{t+1} = Z_Y^{t+1} \Rightarrow y^{t+1} \in Z_Y^{t+1}$

where given a set $S$ and a vector $s'$ we define the addition of them by $S + s' := \{s + s' : s \in S\}$.

Therefore if $x^0 = \mathbf{0} \in Z_X^0$ and $y^0 = \mathbf{0} \in Z_X^0$ then by induction $x_1^t = 0$ for $t < n$.

# C  Proof results from Section 2.4

We will use the following fact throughout the proofs.

**Fact 1.** *Let $(s, \theta) \in S \times [0, 1]^n$ and $z = \text{FORWARD}(s, \theta)$. If Assumption 3 holds then $\mu_i \leq \eta_i$.*

As discussed in Section 2.3, this follows immediately from Theorem 1.

## C.1  Proof of Lemma 1

Lemma 3 is a standard result on the relationship between progress made by a gradient step and the gradient of a function.

**Lemma 3.** *Suppose $s \in S$. If Assumption 2 holds then $\sup_{\hat{s} \in S} \boldsymbol{\nabla}_s \psi_0 \cdot (s - \hat{s}) \leq D_s \sqrt{2L\delta_L}$.*

*Proof.* Note that for any $\hat{s} \in S$,

$$\delta_L(s, \theta) \geq -\underset{\alpha \in \mathbf{R}}{\text{minimize}} \, \alpha \boldsymbol{\nabla}_s \psi_0 \cdot (\hat{s} - s) + \frac{L}{2} \|\hat{s} - s\|_2^2 \alpha^2 = \frac{(\boldsymbol{\nabla}_s \psi_0 \cdot (s - \hat{s}))^2}{2L \|\hat{s} - s\|_2^2}$$

where the equality uses that $\alpha = \frac{\boldsymbol{\nabla}_s \psi_0 \cdot (s - \hat{s})}{L \|\hat{s} - s\|_2^2}$ minimizes the quadratic. Rearranging yields,

$$\boldsymbol{\nabla}_s \psi_0 \cdot (s - \hat{s}) \leq \|\hat{s} - s\|_2 \sqrt{2L \delta_L(s, \theta)}.$$

$\square$

**Lemma 4.** *Suppose Assumption 2 holds, then*

$$\sup_{(\hat{s}, \hat{z}) \in S \times Z} \boldsymbol{\nabla}_{s,z} \mathcal{L} \cdot (s - \hat{s}, z - \hat{z}) \leq D_s \sqrt{2L \delta_L} + D_s \|\boldsymbol{\nabla}_s \mathcal{L} - \boldsymbol{\nabla}_s \psi\|_2 + D_z \|\boldsymbol{\nabla}_z \mathcal{L}\|_2.$$

*Proof.* Moreover,

$$\sup_{(\hat{s}, \hat{z}) \in S \times Z} \boldsymbol{\nabla}_{s,z} \mathcal{L} \cdot (s - \hat{s}, z - \hat{z})$$

$$= \sup_{\hat{s} \in S} \boldsymbol{\nabla}_s \mathcal{L} \cdot (s - \hat{s}) + \sup_{\hat{z} \in Z} \boldsymbol{\nabla}_z \mathcal{L} \cdot (z - \hat{z})$$

$$\leq \sup_{\hat{s} \in S} \boldsymbol{\nabla} \psi_s \cdot (s - \hat{s}) + \sup_{\hat{s} \in S} (\boldsymbol{\nabla}_s \mathcal{L} - \boldsymbol{\nabla}_s \psi) \cdot (s - \hat{s}) + \sup_{z \in Z} \boldsymbol{\nabla}_z \mathcal{L} \cdot (z - \hat{z})$$

$$\leq D_s \sqrt{2L \delta_L} + D_s \|\boldsymbol{\nabla}_s \mathcal{L} - \boldsymbol{\nabla}_s \psi\|_2 + D_z \|\boldsymbol{\nabla}_z \mathcal{L}\|_2$$

where the final inequality uses $\sup_{\hat{s} \in S} \boldsymbol{\nabla} \psi_s \cdot (s - \hat{s}) \leq D_s \sqrt{2L \delta_L}$ (Lemma 3) and Assumption 2. $\square$

Display (8) establishes that $\Delta(s, \theta)$ is a valid duality gap. Moreover, Lemma 4 shows that to provide an upper bound on $\Delta(s, \theta)$ it will suffice to upper bound

$$\sum_{i=1}^{n} (y_i z_i - y_i^+ \mu_i + y_i^- \eta_i) + D_s \sqrt{2L \delta_L} + D_s \|\boldsymbol{\nabla}_s \mathcal{L} - \boldsymbol{\nabla}_s \psi\|_2 + D_z \|\boldsymbol{\nabla}_z \mathcal{L}\|_2.$$

Lemma 5 is our first step towards bounding these quantities. Before proceeding with Lemma 5 we prove a fact we will find useful.

**Fact 2.** *Let $\gamma, t, \alpha, \beta \in \mathbf{R}$ then*

$$(\gamma^+ \beta - \gamma^- \alpha) - \gamma(t\alpha + (1 - t)\beta) = (\beta - \alpha)(\gamma^+ t + \gamma^-(1 - t)).$$

*Proof.* Observe that

$$\gamma^+ (\beta - (t\alpha + (1 - t)\beta)) = \gamma^+ t(\beta - \alpha)$$

$$\gamma^- (-\alpha + (t\alpha + (1 - t)\beta)) = \gamma^-(1 - t)(\beta - \alpha),$$

adding the two expressions together gives the result. $\square$

**Lemma 5.** *Suppose Assumption 4 holds. Let $y_i = \frac{\partial \psi_i}{\partial z_i}$ and $r_i = \theta_i y_i^+ + (1 - \theta_i) y_i^-$, then*

$$\|\boldsymbol{\nabla}_s \mathcal{L} - \boldsymbol{\nabla}_s \psi_0\|_2 \leq \|\boldsymbol{\nabla}_s \mu - \boldsymbol{\nabla}_s \eta\|_2 \|r\|_2 \tag{17a}$$

$$\|\boldsymbol{\nabla}_z \mathcal{L}\|_2 \leq \|\boldsymbol{\nabla}_z \mu - \boldsymbol{\nabla}_z \eta\|_2 \|r\|_2 \tag{17b}$$

$$\sum_{i=1}^{n} (y_i z_i - y_i^+ \mu_i + y_i^- \eta_i) \leq \|\eta - \mu\|_2 \|r\|_2. \tag{17c}$$

*Proof.* Consider the expansion of (6a) and (6b) using $\psi_n = f$:

$$\boldsymbol{\nabla}_s \psi_i = \boldsymbol{\nabla}_s f + \sum_{j=i+1}^{n} \frac{\partial \psi_j}{\partial z_j} (\theta_j \boldsymbol{\nabla}_s \eta_j + (1 - \theta_j) \boldsymbol{\nabla}_s \mu_j)$$

$$\frac{\partial \psi_i}{\partial z_k} = \frac{\partial f}{\partial z_k} + \sum_{j=i+1}^{n} \frac{\partial \psi_j}{\partial z_j} \left( \theta_j \frac{\partial \eta_j}{\partial z_k} + (1 - \theta_j) \frac{\partial \mu_j}{\partial z_k} \right)$$

for each $i \in \{1, \ldots, n\}$ and $k \in \{1, \ldots, i\}$. Setting $k = i$ gives

$$\boldsymbol{\nabla}_s \psi_0 = \boldsymbol{\nabla}_s f + \sum_{j=1}^n \frac{\partial \psi_j}{\partial z_j} \left(\theta_j \boldsymbol{\nabla}_s \eta_j + (1 - \theta_j) \boldsymbol{\nabla}_s \mu_j \right) \tag{18a}$$

$$\frac{\partial \psi_i}{\partial z_i} = \frac{\partial f}{\partial z_i} + \sum_{j=i+1}^n \frac{\partial \psi_j}{\partial z_j} \left(\theta_j \frac{\partial \eta_j}{\partial z_i} + (1 - \theta_j) \frac{\partial \mu_j}{\partial z_i}\right). \tag{18b}$$

Contrast (18) with

$$\boldsymbol{\nabla}_s \mathcal{L} = \boldsymbol{\nabla}_s f + \sum_{j=1}^n \left(y_j^+ \boldsymbol{\nabla}_s \mu_j - y_j^- \boldsymbol{\nabla}_s \eta_j\right) \tag{19a}$$

$$\frac{\partial \mathcal{L}}{\partial z_i} = \frac{\partial f}{\partial z_i} - y_i + \sum_{j=i+1}^n \left(y_j^+ \frac{\partial \mu_j}{\partial z_i} - y_j^- \frac{\partial \eta_j}{\partial z_i}\right). \tag{19b}$$

One can see (18) and (19) share a very similar structure which we will exploit. In particular,

$$\boldsymbol{\nabla}_s \mathcal{L} - \boldsymbol{\nabla}_s \psi_0 = \sum_{i=1}^n \left(y_j^+ \boldsymbol{\nabla}_s \mu_j - y_j^- \boldsymbol{\nabla}_s \eta_j - y_j \left(\theta_j \boldsymbol{\nabla}_s \eta_j + (1 - \theta_j)\boldsymbol{\nabla}_s \mu_j\right)\right)$$

$$= \sum_{j=1}^n \left(\boldsymbol{\nabla}_s \mu_j - \boldsymbol{\nabla}_s \eta_j\right) \left(\theta_j y_j^+ + (1 - \theta_j) y_j^-\right).$$

where the first equality subtracts (18a) from (19a), and the second equality uses Fact 2. We conclude (17a) holds. Similarly,

$$\frac{\partial \mathcal{L}}{\partial z_i} = \sum_{j=i+1}^n \left(y_j^+ \frac{\partial \mu_j}{\partial z_i} - y_j^- \frac{\partial \eta_j}{\partial z_i} - y_j \left(\theta_j \frac{\partial \eta_j}{\partial z_i} + (1 - \theta_j)\frac{\partial \mu_j}{\partial z_i}\right)\right)$$

$$= \sum_{j=i+1}^n \left(\frac{\partial \mu_j}{\partial z_i} - \frac{\partial \eta_j}{\partial z_i}\right) \left(\theta_j y_j^+ + (1 - \theta_j) y_j^-\right).$$

where the first equality substitutes $y_i = \frac{\partial \psi_i}{\partial z_i}$ into (19b) and then subtracts (18b) from (19b), and the second equality uses Fact 2. We conclude (17b) holds. Finally, $z_i - \mu_i = (1 - \theta_i)\mu_i + \theta_i \eta_i - \mu_i = \theta_i(\eta_i - \mu_i)$ and $\eta_i - z_i = (1 - \theta_i)(\eta_i - \mu_i)$ which implies

$$\sum_{i=1}^n y_i^+(z_i - \mu_i) + y_i^-(\eta_i - z_i) = \sum_{i=1}^n (\eta_i - \mu_i)(y_i^+ \theta_i + y_i^-(1 - \theta_i)) = \sum_{i=1}^n r_i(\eta_i - \mu_i),$$

establishing (17c). $\qquad\square$

**Lemma 6.** *Suppose Assumption 2 holds. Let $y_i = \frac{\partial \psi_i}{\partial z_i}$ and $r_i = \theta_i y_i^+ + (1 - \theta_i)y_i^-$, then $\Delta(s, \theta) \le D_s \sqrt{2L\delta_L} + (\|\eta - \mu\|_2 + D_s\|\boldsymbol{\nabla}_s\mu - \boldsymbol{\nabla}_s\eta\|_2 + D_z\|\boldsymbol{\nabla}_z\mu - \boldsymbol{\nabla}_z\eta\|_2) \|r\|_2$.*

*Proof.* Note that by (8), Lemma 4 and 5,

$$\Delta(s, \theta) \le \sum_{i=1}^n (y_i z_i - y_i^+ \mu_i + y_i^- \eta_i) + D_s\sqrt{2L\delta_L} + D_s\|\boldsymbol{\nabla}_s\mathcal{L} - \boldsymbol{\nabla}_s\psi\|_2 + D_z\|\boldsymbol{\nabla}_z\mathcal{L}\|_2$$

$$\le \left(\|\eta - \mu\|_2\|r\|_2 + D_s\left(\sqrt{2L\delta_L} + \|\boldsymbol{\nabla}_s\mu - \boldsymbol{\nabla}_s\eta\|_2\|r\|_2\right) + D_z\|\boldsymbol{\nabla}_z\mu - \boldsymbol{\nabla}_z\eta\|_2\|r\|_2\right).$$

$\qquad\square$

We will find Lemma 6 useful later in Section D.

Next, define

$$t_{i,L} := -\underset{\theta_i + d_i \in [0,1]}{\text{minimize}} \frac{\partial \psi_0}{\partial \theta_i} d_i + \frac{L}{2} d_i^2$$

which represent the guaranteed reduction from a gradient step, contributed by $\theta_i$, assuming the function $\psi_0$ is $L$-smooth.

While Lemma 6 represents useful progress in bounding $\Delta(s, \theta)$. We would like our final bound on $\Delta(s, \theta)$ to depend only on $\delta_L$ and problem constants. Lemma 7 allows us to do that.

**Lemma 7.** *Suppose $\eta_i - \mu_i > 0$. Let $y_i = \frac{\partial \psi_i}{\partial z_i}$ and $r_i = \theta_i y_i^+ + (1 - \theta_i) y_i^-$, then*

$$r_i \leq \frac{\max\left\{\sqrt{2Lt_{i,L}}, 2t_{i,L}\right\}}{\eta_i - \mu_i}. \tag{20}$$

*Proof.* Suppose $\frac{\partial \psi_0}{\partial \theta_i} \geq 0$ and $\theta_i \leq \frac{1}{L} \frac{\partial \psi_0}{\partial \theta_i}$ then

$$t_{i,L} \geq \theta_i \left(\frac{\partial \psi_0}{\partial \theta_i} - \frac{L\theta_i}{2}\right) \geq \frac{\theta_i}{2} \frac{\partial \psi_0}{\partial \theta_i}.$$

If $\frac{\partial \psi_0}{\partial \theta_i} \geq 0$ and $\theta_i \geq \frac{1}{L} \frac{\partial \psi_0}{\partial \theta_i}$ then $t_{i,L} \geq \frac{1}{2L} \left(\frac{\partial \psi_0}{\partial \theta_i}\right)^2$. Therefore, if $\frac{\partial \psi_0}{\partial \theta_i} \geq 0$ then

$$t_{i,L} \geq \frac{1}{2} \min\left\{\frac{1}{L} \left(\frac{\partial \psi_0}{\partial \theta_i}\right)^2, \theta_i \frac{\partial \psi_0}{\partial \theta_i}\right\} \tag{21}$$

which implies that

$$(\eta_i - \mu_i) \frac{\partial \psi_i}{\partial z_i} \theta_i = \frac{\partial \psi_0}{\partial \theta_i} \theta_i \leq \max\left\{\theta_i \sqrt{2Lt_{i,L}}, 2t_{i,L}\right\} \tag{22}$$

where the equality uses (6c) and the inequality rearranges (21). By the same argument, if $\frac{\partial \psi_0}{\partial \theta_i} \leq 0$ then

$$-(\eta_i - \mu_i) \frac{\partial \psi_i}{\partial z_i} (1 - \theta_i) \leq \max\left\{(1 - \theta_i)\sqrt{2Lt_{i,L}}, 2t_{i,L}\right\}. \tag{23}$$

By (22) and (23) we deduce (20). $\qquad\square$

*Proof of Lemma 1.* Observe that

$$\gamma^2 \|r\|_2^2 \leq \sum_{i=1}^n \max\left\{2Lt_{i,L}, 4t_{i,L}^2\right\} \leq \sum_{i=1}^n 2Lt_{i,L} + 4t_{i,L}^2 \leq 2L\delta_L + 4\delta_L^2 \leq 4L\delta_L \tag{24}$$

where the first inequality uses (20), the second inequality uses that $\sum_{i=1}^n t_{i,L} \leq \delta_L$ and last inequality uses the assumption that $\delta_L \leq L/2$. Combining equation (24), Lemma 6 and Assumption 4 gives the result. $\qquad\square$

## C.2 Proof of Theorem 2

Rather than directly prove the Theorem 2, we first prove Lemma 8 which is a generic statement on the convergence of algorithms to minimizers. We will find Lemma 8 useful later in Section D.

**Lemma 8.** *Let $\zeta_1, \zeta_2, \zeta_3 \in (0, \infty)$. Consider a sequence $(s^k, \theta^k)_{k=0}^\infty$ satisfying*

$$\left(f(s^k, \theta^k) - f_*\right)^{\zeta_1 + 1} \leq \zeta_2 (f(s^k, \theta^k) - f(s^{k+1}, \theta^{k+1})) \tag{25}$$

*for all $\frac{f(s^1, \theta^1) - f_*}{f(s^k, \theta^k) - f(s^{k+1}, \theta^{k+1})} \leq \zeta_3$. Then for $K > \zeta_3$*

$$f(s^k, \theta^k) - f_* \leq \left(\frac{\zeta_2}{K - \zeta_3}\right)^{1/\zeta_1}.$$

*Proof.* Define, $f(s^k, \theta^k) - f_* = v^k$. First consider the case that $\frac{f(s^k, \theta^k) - f(s^{k+1}, \theta^{k+1})}{f(s^1, \theta^1) - f_*} \leq \zeta_3$, then by (25) and $v^k - v^{k+1} = f(s^k, \theta^k) - f(s^{k+1}, \theta^{k+1})$ we deduce

$$\frac{(v^k)^{\zeta_1 + 1}}{\zeta_2} \leq v^k - v^{k+1} \Rightarrow v^{k+1} \leq v^k \left(1 - \frac{(v^k)^{\zeta_1}}{\zeta_2}\right).$$

Dividing both sides by $v^{k+1}(v^k)^{\zeta_1}$ and using that $v^{k+1} \le v^k$ yields

$$\frac{1}{(v^k)^{\zeta_1}} \le \frac{v^k}{v^{k+1}} \left( \frac{1}{(v^k)^{\zeta_1-1}} - \frac{1}{\zeta_2} \right) \le \frac{1}{(v^{k+1})^{\zeta_1}} - \frac{1}{\zeta_2}. \tag{26}$$

Furthermore, if $\frac{f(s^k,\theta^k)-f(s^{k+1},\theta^{k+1})}{f(s^1,\theta^1)-f_*} \ge \zeta_3$ then

$$v^{k+1} \le v^k - \frac{f(s^1,\theta^1) - f_*}{\zeta_3} \tag{27}$$

and this can happen at most $\zeta_3$ times. Therefore if $K > \zeta_3$ then

$$\frac{1}{(v^K)^{\zeta_1}} \ge \frac{1}{(v^K)^{\zeta_1}} - \frac{1}{(v^1)^{\zeta_1}} = \sum_{k=1}^{K-1} \frac{1}{(v^{k+1})^{\zeta_1}} - \frac{1}{(v^k)^{\zeta_1}} \ge \frac{K - \zeta_3}{\zeta_2}$$

where the first inequality uses that $v_1 \ge 0$, and second inequality uses (26). Rearranging gives the result. $\qquad \square$

*Proof of Theorem 2.* Define

$$\zeta_1 = 1$$
$$\zeta_2 = L \left( D_s \sqrt{2} + 2\frac{c}{\gamma} \right)^2$$
$$\zeta_3 = \frac{2\Delta(s^1,\theta^1)}{L}.$$

Lemma 1 shows that if $\delta_L(s^k,\theta^k) \le \Delta(s^1,\theta^1)/\zeta_3$ then $\Delta(s^k,\theta^k)^2 \le \zeta_2\delta_L(s^k,\theta^k)$. Furthermore, $\delta_L(s^k,\theta^k) \le \psi_0(s^k,\theta^k) - \psi_0(s^{k+1},\theta^{k+1}) = f(s^k,z^k) - f(s^{k+1},z^{k+1})$ by line 10 of Algorithm 1 and the definition of $\psi_0$ respectively. Combining these two inequalities yields $(f(s^k,z^k) - f_*)^2 \le \zeta_2(f(s^k,z^k) - f(s^{k+1},z^{k+1}))$. Applying Lemma 8 to the latter inequality yields the result. $\qquad \square$

# D   Escaping the basin of a local minimizer

For this section, we consider an alternative set to $\mathcal{K}_\gamma$:

$$\mathcal{C}(s,\theta,q) := \left\{ i : \theta_i \left( \frac{\partial \psi_i}{\partial z_i} \right)^+ + (1 - \theta_i) \left( \frac{\partial \psi_i}{\partial z_i} \right)^- > 2q(\eta_i - \mu_i) \right\}$$

this set identifies the indices where the degeneracy could cause a convergence issue. We then modify SIMPLE-PSI-MINIMIZATION (yielding SAFE-PSI-MINIMIZATION) to detect when the set $C(s,\theta,q)$ is empty and take appropriate action, i.e., call FIX-DEGENERACY. FIX-DEGENERACY extends ESCAPE-EXACT-LOCAL-MIN to allow us to escape from the basin of a local minimizer. To see this, note that if $(s,\theta)$ is fixed then setting $q$ sufficiently large will cause FIX-DEGENERACY$(s,\theta,q)$ to reduce to ESCAPE-EXACT-LOCAL-MIN$(s,\theta)$. However, the value of $q$ needed to achieve this could be arbitrarily large.

**Algorithm 2** Local search algorithm that will find the global minimizer of $\psi_0$.

1: **function** SAFE-PSI-MINIMIZATION($s^1, \theta^1, L, q^1, \epsilon$)
2:     **for** $k = 1, \ldots, \infty$ **do**
3:         *Take corrective action if degeneracy is an issue:*
4:         **if** $\mathcal{C}(s^k, \theta^k, q^k) \neq \emptyset$ **then**
5:             $(s^k, \hat{\theta}^k, \textbf{status}) \leftarrow$ FIX-DEGENERACY($s^k, \theta^k, q^k$)
6:             **if status** = FAILURE **then**
7:                 $q^{k+1} \leftarrow 10q^k$
8:             **end if**
9:         **else**
10:             $\hat{\theta}^k \leftarrow \theta^k$
11:         **end if**
12:         *Termination checks:*
13:         **if** $\Delta(s^k, \hat{\theta}^k) \leq \epsilon$ **then**
14:             **return** $(s^k, \theta^k)$
15:         **end if**
16:         *Reduce the function at least as much as PGD would:*
17:         $(s^{k+1}, \theta^{k+1}) \in \{(s, \theta) : \psi_0(s, \theta) \leq \psi_0(s^k, \hat{\theta}^k) - \delta_L(s^k, \hat{\theta}^k)\}$
18:     **end for**
19: **end function**

---

**Algorithm 3** Algorithm for fixing convergence issues in degenerate case

1: **function** FIX-DEGENERACY($s, \theta, q$)
2:     $z =$ FORWARD($s, \theta$)
3:     $\hat{\theta} \leftarrow \text{copy}(\theta)$
4:     *The minimum reduction in $\psi_0$ if $q \geq Q$:*
5:     $v \leftarrow 0$
6:
7:     **for** $i = n, \ldots, 1$ **do**
8:         *Approximately compute $\frac{\partial \psi_i(s, z_{1:i}, \hat{\theta}_{i+1:n})}{\partial z_i}$:*
9:         $g_i \leftarrow \frac{\partial f}{\partial z_i} + \sum_{j=i+1}^{n} g_j \left( \hat{\theta}_j \frac{\partial \eta_j}{\partial z_i} + (1 - \hat{\theta}_j) \frac{\partial \mu_j}{\partial z_i} \right)$
10:         *Estimate the distance that $z$ has moved:*
11:         $\omega_i \leftarrow \sum_{j=i+1}^{n} \left| \theta_j - \hat{\theta}_j \right| (\eta_j - \mu_j)$
12:         $p_i \leftarrow g_i^+ \theta_i + g_i^- (1 - \theta_i)$
13:         **if** $p_i > 2q(\omega_i + \eta_i - \mu_i)$ **then**
14:             *Fix degeneracy in index $i$:*
15:             $\hat{\theta}_i \leftarrow \begin{cases} 0 & g_i > 0 \\ 1 & g_i < 0 \end{cases}$
16:             $v \leftarrow v + \frac{1}{2} p_i (\eta_i - \mu_i)$
17:         **end if**
18:     **end for**
19:
20:     $\omega_0 \leftarrow \sum_{j=1}^{n} \left| \theta_j - \hat{\theta}_j \right| (\eta_j - \mu_j)$
21:     **if** $\psi_0(s, \hat{\theta}) > \psi_0(s, \theta) - v$ **then**
22:         **return** $(s, \theta, \text{FAILURE})$
23:     **else if** $\left| g_i - \frac{\partial \psi_i(s, \hat{z}_{1:i}, \hat{\theta}_{i+1:n})}{\partial \hat{z}_i} \right| \leq q\omega_0, \forall i$ **then**
24:         **return** $(s, \hat{\theta}, \text{SUCCESS})$
25:     **else**
26:         **return** $(s, \hat{\theta}, \text{FAILURE})$
27:     **end if**
28: **end function**

**Remark 5.** *With careful implementation the cost of running* FIX-DEGENERACY *is the same as one backpropagation.*

**Remark 6.** *If $\mathcal{C}(s, \theta, q) = \emptyset$ then $(s, \hat{\theta}, \textbf{status}) \leftarrow$ FIX-DEGENERACY$(s, \theta, q)$ satisfies $\hat{\theta} = \theta$ and* **status** $=$ SUCCESS. *In other words, removing Line 4 and Line 9-11 of* SAFE-PSI-MINIMIZATION *does not change the behaviour of the Algorithm (although it may create unnecessary computation).*

This section introduces two new assumptions (Assumption 5 and 6). We defer justifying these assumptions to Section E where we show that Assumptions 1, 2, and 3 imply that these introduced assumptions hold.

**Assumption 5.** *Let $z \leftarrow$ FORWARD$(s, \theta)$,*

$$g_i = \frac{\partial f}{\partial z_i} + \sum_{j=i+1}^{n} g_j \left( \hat{\theta}_j \frac{\partial \eta_j}{\partial z_i} + (1 - \hat{\theta}_j) \frac{\partial \mu_j}{\partial z_i} \right),$$

*and $\hat{z} \leftarrow$ FORWARD$(s, \hat{\theta})$. Then there exists a constant $Q > 0$ such that*

$$\left| g_i - \frac{\partial \psi_i(s, \hat{z}_{1:i}, \hat{\theta}_{i+1:n})}{\partial \hat{z}_i} \right| \leq Q \sum_{i=1}^{n} \left| \theta_i - \hat{\theta}_i \right| \left| \eta_i - \mu_i \right|.$$

**Lemma 9.** *Suppose Assumption 3 and 5 holds. If $q \geq Q$ then $(s, \hat{\theta}, \textbf{status}) \leftarrow$ FIX-DEGENERACY$(s, \theta, q)$ has* **status** $=$ SUCCESS.

*Proof.* By Lines 21 to 27 of FIX-DEGENERACY we can see for **status** $=$ SUCCESS we need both

$$\left| g_i - \frac{\partial \psi_i(s, \hat{z}_{1:i}, \hat{\theta}_{i+1:n})}{\partial \hat{z}_i} \right| \leq q\omega_0, \forall i \qquad (28)$$

and

$$\psi_0(s, \hat{\theta}) \leq \psi_0(s, \theta) - v. \qquad (29)$$

We establish each of these in turn. First observe that (28) holds immediately by Assumption 5 and definition of $\omega_0$ in line 20 of FIX-DEGENERACY.

Next we show (29) by bounding each of the right hand side terms in the equality

$$\psi_0(s, \hat{\theta}) - \psi_0(s, \theta) = \sum_{i=1}^{n} \psi_0(s, \theta_{1:i-1}, \hat{\theta}_{i:n}) - \psi_0(s, \theta_{1:i}, \hat{\theta}_{i+1:n}). \qquad (30)$$

Recall the definition of $p_i$, $\omega_i$, and $\hat{\theta}_i$ from FIX-DEGENERACY. If $p_i > 2q(\omega_i + \eta_i - \mu_i)$ then

$\psi_0(s, \theta_{1:i-1}, \hat{\theta}_{i:n}) - \psi_0(s, \theta_{1:i}, \hat{\theta}_{i+1:n})$

$= \int_{\theta_i}^{\hat{\theta}_i} \frac{\partial \psi_0(s, \theta_{1:i-1}, \gamma, \hat{\theta}_{i+1:n})}{\partial \gamma} \partial \gamma$

$= \int_{\theta_i}^{\hat{\theta}_i} \frac{\partial \psi_i(s, z_{1:i-1}, t, \hat{\theta}_{i+1:n})}{\partial t}(\eta_i - \mu_i)\partial t$ $\hspace{2cm}$ by (6c)

$= \int_{\theta_i}^{\hat{\theta}_i} g_i(\eta_i - \mu_i)\partial t + \int_{\theta_i}^{\hat{\theta}_i} \left( \frac{\partial \psi_i(s, z_{1:i-1}, t, \hat{\theta}_{i+1:n})}{\partial t} - g_i \right)(\eta_i - \mu_i)\partial t$

$\leq g_i(\eta_i - \mu_i) \int_{\theta_i}^{\hat{\theta}_i} \partial t + Q \left( \sum_{j=i+1}^{n} \left| \theta_j - \hat{\theta}_j \right| (\eta_j - \mu_j) \right)(\eta_i - \mu_i) \left| \int_{\theta_i}^{\hat{\theta}_i} \partial t \right|$ $\hspace{1cm}$ by Assumption 5

$= g_i(\eta_i - \mu_i) \int_{\theta_i}^{\hat{\theta}_i} \partial t + Q\omega_i(\eta_i - \mu_i) \left| \int_{\theta_i}^{\hat{\theta}_i} \partial t \right|$ $\hspace{2cm}$ by definition of $\omega_i$

$= g_i(\eta_i - \mu_i)(\hat{\theta}_i - \theta_i) + Q\omega_i(\eta_i - \mu_i) \left| \hat{\theta}_i - \theta_i \right|$

$\leq g_i(\eta_i - \mu_i)(\hat{\theta}_i - \theta_i) + q\omega_i(\eta_i - \mu_i) \left| \hat{\theta}_i - \theta_i \right|$ $\hspace{2cm}$ by $Q \leq q$

$\leq g_i(\eta_i - \mu_i)(\hat{\theta}_i - \theta_i) + q\omega_i(\eta_i - \mu_i)$ $\hspace{2cm}$ as $\theta_i, \hat{\theta}_i \in [0, 1]$

$= (\eta_i - \mu_i)(q\omega_i - p_i)$ $\hspace{2cm}$ by definition of $p_i$ and $\hat{\theta}_i$

$\leq -\frac{p_i(\eta_i - \mu_i)}{2},$ $\hspace{8cm}$ (31)

where the last inequality uses $p_i > 2q\omega_i$.

If $p_i \leq 2q(\omega_i + \eta_i - \mu_i)$ then

$$\psi_0(s, \theta_{1:i-1}, \hat{\theta}_{i:n}) - \psi_0(s, \theta_{1:i}, \hat{\theta}_{i+1:n}) = 0 \hspace{2cm} (32)$$

Therefore, by (30), (31), (32), and definition of $v$ we establish (29). $\hspace{1cm}$ $\square$

**Assumption 6.** *Denote* $z = \text{FORWARD}(s, \theta)$ *and* $\hat{z} = \text{FORWARD}(s, \hat{\theta})$. *There exists a constant* $P > 0$ *such that*

$$|\eta_i - \mu_i - (\hat{\eta}_i - \hat{\mu}_i)| \leq P \sum_{i=1}^{n} \left| \theta_i - \hat{\theta}_i \right| |\eta_i - \mu_i|,$$

$\forall s \in S, \forall \theta, \hat{\theta} \in [0, 1]^n, \forall i \in \{1, \ldots, n\}$ *with* $\hat{\eta}_i := \eta_i(s, \hat{z}_{1:i-1})$ *and* $\hat{\mu}_i := \mu(s, \hat{z}_{1:i-1})$.

**Lemma 10.** *Suppose that Assumption 3 and 6 hold. Let* $(s, \hat{\theta}, \textbf{status}) \leftarrow \text{FIX-DEGENERACY}(s, \theta, q)$, *and* $\hat{z} = \text{FORWARD}(s, \hat{\theta})$. *If* $\textbf{status} = \text{SUCCESS}$ *then*

$$\hat{\theta}_i \left( \frac{\partial \psi_i(s, \hat{z}_{1:i}, \hat{\theta}_{i+1:n})}{\partial \hat{z}_i} \right)^+ + (1 - \hat{\theta}_i) \left( \frac{\partial \psi_i(s, \hat{z}_{1:i}, \hat{\theta}_{i+1:n})}{\partial \hat{z}_i} \right)^- \leq q((3 + P)\omega_0 + \hat{\eta}_i - \hat{\mu}_i)$$
$$\hspace{14cm} (33)$$

*and* $\omega_0 \leq \sqrt{2v/q}$.

*Proof.* Denote $\hat{\psi}_i := \psi_i(s, \hat{z}_{1:i}, \hat{\theta}_{i+1:n})$. From Line 23 of SAFE-PSI-MINIMIZATION

$$\left| g_i - \frac{\partial \hat{\psi}_i}{\partial \hat{z}_i} \right| \leq q\omega_0. \hspace{2cm} (34)$$

Therefore,

$$\hat{\theta}_i \left( \frac{\partial \hat{\psi}_i}{\partial \hat{z}_i} \right)^+ + (1 - \hat{\theta}_i) \left( \frac{\partial \hat{\psi}_i}{\partial \hat{z}_i} \right)^- \leq g_i^+ \hat{\theta}_i + g_i^- (1 - \hat{\theta}_i) + q\omega_0 \hspace{2cm} (35)$$

where the inequality uses (34) and $\hat{\theta}_i \in [0,1]$. From Line 12-15 if $p_i > 2q(\omega_i + \eta_i - \mu_i)$ then

$$g_i^+ \hat{\theta}_i + g_i^- (1 - \hat{\theta}_i) = 0 \Rightarrow \hat{\theta}_i \left( \frac{\partial \hat{\psi}_i}{\partial \hat{z}_i} \right)^+ + (1 - \hat{\theta}_i) \left( \frac{\partial \hat{\psi}_i}{\partial \hat{z}_i} \right)^- \le q\omega_0$$

where the implication uses (35). On the other hand, if

$$g_i^+ \theta_i + g_i^- (1 - \theta_i) = p_i \le 2q(\omega_i + \eta_i - \mu_i)$$

then by $\hat{\theta}_i = \theta_i$ and (35),

$$\hat{\theta}_i \left( \frac{\partial \hat{\psi}_i}{\partial \hat{z}_i} \right)^+ + (1 - \hat{\theta}_i) \left( \frac{\partial \hat{\psi}_i}{\partial \hat{z}_i} \right)^- \le p_i + q\omega_0 = q(3\omega_0 + \eta_i - \mu_i). \tag{36}$$

Furthermore, by Assumption 6

$$\left| \eta_i - \mu_i - (\hat{\eta}_i - \hat{\mu}_i) \right| \le P\omega_0$$

which combined with (36) yields (33).

It remains to show that $\omega_0 \le \sqrt{2v/q}$. Note that

$$\eta_i - \mu_i \ge \left| \theta_i - \hat{\theta}_i \right| (\eta_i - \mu_i) = \omega_{i-1} - \omega_i \tag{37}$$

where the inequality uses $\theta_i, \hat{\theta}_i \in [0,1]$. Define $\mathcal{I} := \{ i \in \{1, \dots, n\} : \theta_i \ne \hat{\theta}_i \}$. Finally,

$$
\begin{aligned}
v &= \frac{1}{2} \sum_{i \in \mathcal{I}} (\eta_i - \mu_i) p_i && \text{by definition of } v \\
&\ge \frac{1}{2} \sum_{i \in \mathcal{I}} (\omega_{i-1} - \omega_i) p_i && \text{by (37)} \\
&> q \sum_{i \in \mathcal{I}} (\omega_{i-1} - \omega_i) \omega_{i-1} && \text{by } p_i > 2q(\omega_i + \eta_i - \mu_i) > 2\omega_i \text{ for } i \in \mathcal{I} \\
&= q \sum_{i=1}^{n} (\omega_{i-1} - \omega_i) \omega_{i-1} && \text{by } \omega_{i-1} - \omega_i \text{ for } i \notin \mathcal{I} \\
&= q \sum_{i=1}^{n} \sum_{j=i}^{n} (\omega_{i-1} - \omega_i)(\omega_{j-1} - \omega_j) && \\
&\ge \frac{q(\omega_0 - \omega_n)^2}{2} && \text{by Fact 3} \\
&= \frac{q\omega_0^2}{2} && \text{since } \omega_n = 0.
\end{aligned}
$$

Rearranging this inequality gives the result. $\square$

**Fact 3.** *Let* $\delta_1, \dots, \delta_n \in \mathbf{R}$ *then*

$$\left( \sum_{i=1}^{n} \delta_i \right)^2 \le 2 \sum_{i=1}^{n} \sum_{j=i}^{n} \delta_i \delta_j.$$

*Proof.* It follows by

$$
\begin{aligned}
\left( \sum_{i=1}^{n} \delta_i \right)^2 &= \sum_{i=1}^{n} \sum_{j=1}^{n} \delta_i \delta_j = \sum_{i=1}^{n} \delta_i^2 + \sum_{i=1}^{n} \sum_{j=i+1}^{n} \delta_i \delta_j + \sum_{i=1}^{n} \sum_{j=1}^{i-1} \delta_i \delta_j \\
&= \sum_{i=1}^{n} \delta_i^2 + 2 \sum_{i=1}^{n} \sum_{j=i+1}^{n} \delta_i \delta_j \\
&= - \sum_{i=1}^{n} \delta_i^2 + 2 \sum_{i=1}^{n} \sum_{j=i}^{n} \delta_i \delta_j.
\end{aligned}
$$

$\square$

**Lemma 11.** *Suppose that Assumption 3 and 6 hold. Let* $(s, \hat{\theta}, \textbf{status}) \leftarrow \textsc{Fix-Degeneracy}(s, \theta, q)$, *and* $\hat{z} = \textsc{Forward}(s, \hat{\theta})$. *If* $\textbf{status} = \textsc{Success}$ *then for all* $i \in \{1, \dots, n\}$,

$$\hat{r}_i \leq \sqrt{q} \left( (3 + P)\sqrt{v} + \max \left\{ \sqrt[4]{2L\hat{t}_{i,L}}, \sqrt{2\hat{t}_{i,L}} \right\} \right).$$

*with* $\hat{y}_i := \frac{\partial \psi_i(s, \hat{z}_{1:i}, \hat{\theta}_{i+1:n})}{\partial \hat{z}_i}$, $\hat{r}_i := \hat{\theta}_i \hat{y}_i^+ + (1 - \hat{\theta}_i)\hat{y}_i^-$, *and* $\hat{t}_{i,L} := - \underset{\hat{\theta}_i + d_i \in [0,1]}{minimize} \frac{\partial \psi_i(s, \hat{z}_{1:i}, \hat{\theta}_{i+1:n})}{\partial \hat{\theta}_i} d_i + \frac{L}{2} d_i^2$.

*Proof.* Define

$$\gamma_i := \begin{cases} \frac{1}{\hat{\eta}_i - \hat{\mu}_i} & \hat{\eta}_i - \hat{\mu}_i \neq 0 \\ \infty & \text{otherwise.} \end{cases}$$

$$a := \max \left\{ \sqrt{2L\hat{t}_{i,L}}, 2\hat{t}_{i,L} \right\}$$

$$b := q(3 + P)\omega_0$$

$$c := q.$$

By Lemma 7, $\hat{r}_i \leq a\gamma_i$ and by Lemma 10, $\hat{r}_i \leq b + c/\gamma_i$. Therefore,

$$\hat{r}_i \leq \min\{a\gamma_i, b + c/\gamma_i\}.$$

Maximizing the upper bound with for $\gamma_i \in [0, \infty]$ gives

$$a\gamma_i = b + c/\gamma_i \Rightarrow a\gamma_i^2 - b\gamma_i - c = 0 \Rightarrow \gamma_i = \frac{b + \sqrt{b^2 + 4ac}}{2a}$$

where the second implication uses the quadratic formula and $\gamma_i \geq 0$. Plugging this back in gives

$$\hat{r}_i \leq \frac{b + \sqrt{b^2 + 4ac}}{2} \Rightarrow \hat{r}_i \leq b + \sqrt{ac}.$$

Therefore, using $\omega_0 \leq \sqrt{2v/q}$ from Lemma 10 and the definition of $a, b, c$ we get

$$\hat{r}_i \leq (3 + P)\sqrt{2vq} + \sqrt{q \max \left\{ \sqrt{2L\hat{t}_{i,L}}, 2\hat{t}_{i,L} \right\}}.$$

$\square$

**Lemma 12.** *Suppose that Assumption 2, 3, 4, and 6 holds. Define* $\tau := \delta_L(s, \hat{\theta}) + v$. *If*

$$\tau \leq \min \left\{ 2L, \frac{2L}{n(3 + P)^4}, \frac{4D_s^4 L^2}{c^4} \right\} \tag{38}$$

*then*

$$\Delta(s, \hat{\theta})^4 \leq 96c^4 q^2 L\tau.$$

*Proof.* First observe that

$$\begin{aligned} \|r\|_2^4 &\leq q^2 \left( (3 + P)^4 nv^2 + 4\delta_L(s, \hat{\theta})^2 + 2L\delta_L(s, \hat{\theta}) \right) \\ &\leq q^2 \left( (3 + P)^4 n\tau^2 + 4\tau^2 + 2L\tau \right) \\ &\leq 6q^2 L\tau \end{aligned} \tag{39}$$

where the first inequality uses Lemma 11, the second inequality uses the definition of $\tau$, and the third inequality uses (38).

Next,

$$\begin{aligned} \Delta(s, \hat{\theta}) &\leq D_s \sqrt{2L\delta_L(s, \hat{\theta})} + c\|r\|_2 \\ &\leq D_s \sqrt{2L\tau} + c\sqrt[4]{6q^2 L\tau} \\ &\leq 2c\sqrt[4]{6q^2 L\tau} \end{aligned}$$

where the first inequality uses Lemma 6 and Assumption 4, the second inequality uses (39), and the third uses (38). $\square$

**Theorem 4.** *Suppose that Assumption 2, 3, 4, 5, and 6 hold. Define*

$$\zeta_2 = 96c^4 q^2 L$$

$$\zeta_3 = \frac{\Delta(s^1, \theta^1)}{\min\left\{ 2L, \frac{2L}{n(3+B)^4}, \frac{4D_s^4 L^2}{c^4} \right\}}$$

*where $L$ is the smoothness constant for $\psi_0$. Then for $k > \zeta_3$* SAFE-PSI-MINIMIZATION *satisfies*

$$\Delta(s^k, \theta^k) \leq \left( \frac{\zeta_2}{k - \zeta_3} \right)^{1/3}$$

*Proof.* Follows by Lemma 12 and Lemma 8 with $\zeta_1 = 3$. $\qquad\square$

### D.1 Proof of Theorem 3

*Proof.* Assumption 1 and 2 imply that $f$, $\eta_i$, and $\mu_i$ are $\beta$-smooth and $B$-Lipschitz for some constant $\beta, B > 0$. Since $\mu_i$ and $\eta_i$ are Lipschitz Assumption 4 holds. Lemma 14 implies Assumption 5 holds. Corollary 1 implies Assumption 6 holds. Note that Corollary 1 and Lemma 14 appear in Section E

With Assumption 5, and 6 established, the result holds by Theorem 4. $\qquad\square$

## E   Justifying assumptions

The purpose of this section is to show that if Assumptions 1 and 2 hold then Assumption 5 and 6 hold.

**Definition 2.** *A function $h : X \to \mathbf{R}$ is $L$-smooth with respect to $\|\cdot\|$ if $\|\nabla h(x) - \nabla h(x')\|_* \leq L\|x - x'\|$ for all $x, x' \in X$.*

**Definition 3.** *A function $h : X \to \mathbf{R}$ is $B$-Lipschitz with respect to $\|\cdot\|$ if $|h(x) - h(x')| \leq B\|x - x'\|$ for all $x, x' \in X$.*

**Fact 4.** *Suppose that $h : X \to \mathbf{R}$ differentiable and $B$-Lipschitz with respect to $\|\cdot\|$, then $\|\nabla h(x)\|_* \leq B$ for all $x \in X$.*

**Fact 5.** *Suppose that the function $h : X \to \mathbf{R}$ is smooth and $X$ is bounded. Then (for any given norm) there exists constant $B$ and $\beta$ such that $h$ is $B$-Lipschitz and $\beta$-smooth.*

### E.1   Proof Assumption 6 holds

**Lemma 13.** *Suppose that $\eta_i - \mu_i$ is $B$-Lipschitz with respect to the $\ell_1$-norm for $B > 0$. Let $z = $ FORWARD$(s, \theta)$ and $\hat{z} = $ FORWARD$(\hat{s}, \hat{\theta})$. Then*

$$\|z - \hat{z}\|_1 \leq B^{-1}(1 + B)^n \sum_{i=1}^{n} \left( \left|\theta_i - \hat{\theta}_i\right| |\eta_i - \mu_i| + \|s - \hat{s}\|_1 \right)$$

*for all $\theta \in [0,1]^n$.*

*Proof.* Denote $\eta_i(s, z_{1:i-1})$, $\mu_i(s, z_{1:i-1})$, $\eta_i(\hat{s}, \hat{z}_{1:i-1})$, $\mu_i(\hat{s}, \hat{z}_{1:i-1})$ by $\eta_i, \mu_i, \hat{\eta}_i$ and $\hat{\mu}_i$ respectively. Observe that

$$\begin{aligned}
|z_i - \hat{z}_i| &= \left| \theta_i \eta_i + (1 - \theta_i)\mu_i - (\hat{\theta}_i \hat{\eta}_i + (1 - \hat{\theta}_i)\hat{\mu}_i) \right| \\
&= \left| (\theta_i - \hat{\theta}_i)(\eta_i - \mu_i) + (1 - \hat{\theta}_i)(\mu_i - \hat{\mu}_i) + \hat{\theta}_i(\eta_i - \hat{\eta}_i) \right| \\
&\leq \left| (\theta_i - \hat{\theta}_i)(\eta_i - \mu_i) \right| + |\mu_i - \hat{\mu}_i + \eta_i - \hat{\eta}_i| \\
&\leq \left| (\theta_i - \hat{\theta}_i)(\eta_i - \mu_i) \right| + B(\|s - \hat{s}\|_1 + \|z_{1:i-1} - \hat{z}_{1:i-1}\|_1). \quad (40)
\end{aligned}$$

Applying (40) for $i = 1$ we have $|z_1 - \hat{z}_1| \leq \left| (\theta_i - \hat{\theta}_i)(\eta_i - \mu_i) \right| + B\|s - \hat{s}\|_1$. Suppose that,

$$\|z_{1:i} - \hat{z}_{1:i}\|_1 \leq \sum_{j=1}^{i} (B+1)^{i-j} \left( B\|s - \hat{s}\|_1 + \left| (\theta_j - \hat{\theta}_j)(\eta_j - \mu_j) \right| \right)$$

then by (40) we deduce

$$\|z_{1:i+1} - \hat{z}_{1:i+1}\|_1 \le \sum_{j=1}^{i+1}(B+1)^{1+i-j}\left(B\|s-\hat{s}\|_1 + \left|(\theta_j - \hat{\theta}_j)(\eta_j - \mu_j)\right|\right).$$

By induction and the fact $\sum_{j=1}^{n}(1+B)^{n-j} \le B(B+1)^n$, the result holds. $\qquad\square$

**Corollary 1.** *Suppose that $\eta_i - \mu_i$ is B-Lipschitz with respect to the $\ell_1$-norm for $B > 0$. Let $z = \text{FORWARD}(s, \theta)$ and $\hat{z} = \text{FORWARD}(\hat{s}, \hat{\theta})$. Then*

$$|\eta_i - \mu_i - (\eta_i(\hat{s}, \hat{z}_{1:i-1}) - \mu_i(\hat{s}, \hat{z}_{1:i-1}))| \le \beta\|s-\hat{s}\|_1 + \beta B^{-1}(1+B)^n \sum_{i=1}^{n}\left(\left|\theta_i - \hat{\theta}_i\right||\eta_i - \mu_i| + \|s - \hat{s}\|_1\right)$$

*for all $\theta \in [0,1]^n$.*

*Proof.* Follows from the assumption $\eta_i - \mu_i$ is $\beta$-smooth and Lemma 13. $\qquad\square$

Therefore, if Assumptions 1 and 2 hold then, by Fact 5 and Corollary 1, we conclude Assumption 6 holds.

### E.2  Proof Assumption 5 holds

**Fact 6.** *Let $a \in \mathbf{R}^n, b \in \mathbf{R}^{n+1}$. Suppose*

$$a_i := b_{n+1} + \sum_{j=i+1}^{n} a_j b_j,$$

*then $\|a\|_1 \le (1 + \|b\|_\infty)^n$.*

*Proof.* Note that

$$|a_i| \le \|b\|_\infty + \|a_{i+1:n}\|_1 \|b\|_\infty \Rightarrow \|a_{i:n}\|_1 \le \|b\|_\infty + \|a_{i+1:n}\|_1(1 + \|b\|_\infty).$$

Therefore, if

$$\|a_{i:n}\|_1 \le \sum_{j=1}^{n-i} \|b\|_\infty (1 + \|b\|_\infty)^j. \tag{41}$$

then

$$\|a_{i-1:n}\|_1 \le \|b\|_\infty + \sum_{j=1}^{n-i} \|b\|_\infty (1 + \|b\|_\infty)^{j+1} = \sum_{j=1}^{n-(i-1)} \|b\|_\infty (1 + \|b\|_\infty)^j.$$

Since (41) holds for $i = n$ by induction (41) holds for all $i$. By the bound on the sum of a geometric series the result holds. $\qquad\square$

**Fact 7.** *Let $a, c \in \mathbf{R}^n, b \in \mathbf{R}^{n\times n}$. Suppose*

$$a_i := c_i + \sum_{j=i+1}^{n} a_j b_{i,j}, \quad \hat{a}_i := \hat{c}_i + \sum_{j=i+1}^{n} \hat{a}_j \hat{b}_{i,j},$$

*then*

$$\|a - \hat{a}\|_1 \le \left(\|c - \hat{c}\|_\infty + (1 + \|\hat{b}\|_\infty)^n \|\hat{b} - b\|_\infty\right) \frac{(1 + \|b\|_\infty)^n}{\|b\|_\infty}.$$

*Proof.* Note that

$$|a_i - \hat{a}_i| = \left|c_i - \hat{c}_i + \sum_{j=i+1}^{n}(a_j b_{i,j} - \hat{a}_j \hat{b}_{i,j})\right|$$

$$= \left|c_i - \hat{c}_i + \sum_{j=i+1}^{n}((a_j - \hat{a}_j)b_j - \hat{a}_j(\hat{b}_{i,j} - b_{i,j}))\right|$$

$$\le \|a_{i+1:n} - \hat{a}_{i+1:n}\|_1 \|b\|_\infty + \|c - \hat{c}\|_\infty + \|\hat{a}\|_1 \|\hat{b} - b\|_\infty$$

$$\le \|a_{i+1:n} - \hat{a}_{i+1:n}\|_1 \|b\|_\infty + \|c - \hat{c}\|_\infty + (1 + \|\hat{b}\|_\infty)^n \|\hat{b} - b\|_\infty$$

where the last inequality uses Fact 6. By induction,

$$\|a_{i:n} - \hat{a}_{i:n}\|_1 \leq \left( \|c - \hat{c}\|_\infty + (1 + \|\hat{b}\|_\infty)^n \|\hat{b} - b\|_\infty \right) \sum_{j=1}^{n-i} (1 + \|b\|_\infty)^j.$$

By the bound on the sum of a geometric series the result holds. □

**Lemma 14.** *Suppose $f$, $\eta_i$, and $\mu_i$ are $\beta$-smooth functions with respect to the $\ell_1$-norm. Also, suppose $f$, $\eta_i$ and $\mu_i$ are $B$-Lipschitz with respect to the $\ell_1$-norm and $\hat{\theta}, \theta \in [0,1]^n$. Let $z \leftarrow \mathrm{FORWARD}(s, \theta)$,*

$$g_i := \frac{\partial f}{\partial z_i} + \sum_{j=i+1}^{n} g_j \left( \hat{\theta}_j \frac{\partial \eta_j}{\partial z_i} + (1 - \hat{\theta}_j) \frac{\partial \mu_j}{\partial z_i} \right),$$

*and $\hat{z} \leftarrow \mathrm{FORWARD}(s, \hat{\theta})$. Then there exists $Q \leq 2\beta B^{-2}(1 + B)^{3n}$ s.t.*

$$\sum_{i=1}^{n} \left| g_i - \frac{\partial \psi_i(s, \hat{z}_{1:i-1}, \hat{\theta}_{i+1:n})}{\partial \hat{z}_i} \right| \leq Q \sum_{i=1}^{n} \left| \theta_i - \hat{\theta}_i \right| |\eta_i - \mu_i|.$$

*Proof.* Denote $\eta_i(s, z_{1:i-1})$, $\mu_i(s, z_{1:i-1})$, $\eta_i(s, \hat{z}_{1:i-1})$, $\mu_i(s, \hat{z}_{1:i-1})$, $\psi_i(s, \hat{z}_{1:i-1}, \hat{\theta}_{i+1:n})$ by $\eta_i, \mu_i, \hat{\eta}_i, \hat{\mu}_i$, and $\hat{\psi}_i$ respectively. This proof is based on Fact 7. Define,

$$a_i = \frac{\partial \psi_i}{\partial z_i}, \quad b_{i,j} = \hat{\theta}_j \frac{\partial \eta_j}{\partial z_i} + (1 - \hat{\theta}_j) \frac{\partial \mu_j}{\partial z_i}, \quad c_i = \frac{\partial f}{\partial z_j}$$

$$\hat{a}_i = \frac{\partial \hat{\psi}_i}{\partial \hat{z}_i}, \quad \hat{b}_{i,j} = \hat{\theta}_j \frac{\partial \hat{\eta}_j}{\partial z_i} + (1 - \hat{\theta}_j) \frac{\partial \hat{\mu}_j}{\partial z_i}, \quad \hat{c}_i = \frac{\partial \hat{f}}{\partial z_j}.$$

Moreover, observe that by our assumptions

$$\left| b_{i,j} - \hat{b}_{i,j} \right| \leq \hat{\theta}_j \left| \frac{\partial \eta_j}{\partial z_i} - \frac{\partial \hat{\eta}_j}{\partial \hat{z}_i} \right| + (1 - \hat{\theta}_j) \left| \frac{\partial \mu_j}{\partial z_i} - \frac{\partial \hat{\mu}_j}{\partial \hat{z}_i} \right| \leq \beta \|z - \hat{z}\|_1 \qquad (42\text{a})$$

$$\|c - \hat{c}\|_\infty \leq \beta \|z - \hat{z}\|_1 \qquad (42\text{b})$$

$$\max\{\|b\|_\infty, \|\hat{b}\|_\infty\} \leq B. \qquad (42\text{c})$$

Therefore,

$$\|a - \hat{a}\|_1 \leq (\beta \|z - \hat{z}\|_1 + (1 + B)^n \beta \|z - \hat{z}\|_1) \frac{(1 + B)^n}{B}$$

$$\leq 2\beta B^{-1}(1 + B)^{2n} \|z - \hat{z}\|_1$$

$$\leq 2\beta B^{-2}(1 + B)^{3n} \sum_{i=1}^{n} \left| \theta_i - \hat{\theta}_i \right| |\eta_i - \mu_i|$$

where the first inequality uses (42) and Fact 7, the second inequality uses $(1 + B)^n \geq 1$, and the third uses Lemma 13. □

Therefore, if Assumptions 1 and 2 hold then, by Fact 5 and Lemma 14, we conclude Assumption 5 holds.

## F Implementation details

The fastest known way to derive bounds on the values taken by neural network is Interval Bound Propagation (IBP) [25]. It does not attempt to solve the optimization problem of the type of (1) which we define in Appendix A.2 but simply performs interval analysis to derive loose bounds on the values taken by the neural network. As a result, running IBP is a very fast procedure, whose cost is analogous to performing a forward pass through the network, but will give much looser bounds than solving the optimization problem. We include it in Table 1 as a lower bound on the runtime that can be achieved by any of the method. We also use it to derive the intermediate bounds $l_i$ and $u_i$ required for the definition of the constraints (14a) when formulating the optimization problems for the methods we are benchmarking.

For DeepVerify [6], we use the code released by the authors. The Lagrangian dual is optimized using a sub-gradient method. We use the Adam optimizer [46], starting with an initial step size of $10^{-2}$ and decreasing it by a factor of 10 every 100 steps. We run the algorithm for a maximum of 500 steps, but allow early stopping if the dual objective value reaches the value returned by the dual objective of our NonConvex reformulation. In its original formulation, DeepVerify does not have access to a primal objective and can therefore not be provided with an appropriate stopping criterion. We provide it here with one in order to generate a more fair comparison,

Our NonConvex relaxation is solved following the principle of SIMPLE-PSI-MINIMIZATION. To optimize the objective function (line 10 of Algorithm 1), we use the FISTA with backtracking linesearch [19] algorithm. We initialize our step size to a high value of 100, and at each optimization step, perform a line-search starting from the previous step-size, progressively shrinking it by a backtracking coefficient of 0.8 until sufficient progress can be guaranteed. In order to prevent the use of too small learning rates, we increase the step-size by a factor of 1.5 when no backtracking was necessary, and clip the step size to a minimum of $10^{-5}$. We run a maximum of 50 steps of this algorithm, unless we first reach a situation where the relative gap between the primal and the dual objective becomes smaller than $10^{-2}$.

For the CVXPY [28] based solvers, we used the default settings.

## G  Benchmarking on Robustly trained Networks

In addition to the results reported in Section 3, we provide additional results for networks trained with Interval Bound Propagation (IBP) [25], to achieve robustness against $\mathcal{L}_\infty$-bounded perturbations of size $\epsilon = 8.0/255$. Because of the strong regularization effect of IBP, there is very little difference between the solution to the optimization problem described in Section A.2 and the bounds produced by IBP, so better optimization algorithms do not prove as useful.

For these experiments, we didn't employ the relative duality gap criterion and instead optimized until the duality gap reached a value smaller than 1e-4.

Even on those networks for which the solution should be easier to obtain, we observe similar results. Off-the-shelf solvers based on CVXPY do not scale beyond the Tiny network, and the runtime for DeepVerify is significantly larger than for our nonconvex solver, as shown in Table 3. The discrepancy in runtime comes from the fact that DeepVerify does not manage to converge to an accurate enough solution, as shown by the low percentage of early stopping in Table 4 and the observable plateaus in Figure 7a.

We note that as opposed to networks trained with adversarial training, Figure 7b reveals that when computing bounds on networks trained to achieve certifiable robustness, activations do get close to potentially degenerate points, which might be problematic for the optimization procedure and require us to employ Algorithm 3 to find optimal solutions.

| Methods | ReLU activation | | | SoftPlus activation | | |
|---|---|---|---|---|---|---|
| | Tiny | Small | Medium | Tiny | Small | Medium |
| IBP [25] | 3.4 | 2.8 | 3.2 | 3.9 | 2.3 | 2.8 |
| DeepVerify [6] | 338 | 658 | 1171 | 403 | 842 | 1620 |
| NonConvex (ours) | 93 | 176 | 297 | 6.1 | 61 | 210 |
| Solver (SCS) [26] | $78.4 \cdot 10^3$ | - | - | $32.1 \cdot 10^4$ | - | - |
| Solver (ECOS) [27] | $41.2 \cdot 10^3$ | - | - | - | - | - |

Table 3: Runtime of bound computation, in milliseconds. Results correspond to the computation of a lower bound on the gap between ground truth and all other classes for a network subject to an $L_\infty$ adversarial attack with budget $\frac{8.0}{255}$, averaged over the CIFAR-10 test set, on network trained with IBP.

| | Proportion of early stopping | | Average number of iterations | |
|---|---|---|---|---|
| | DeepVerify | NonConvex | DeepVerify | NonConvex |
| Tiny ReLU | 92% | 99% | 242 | 6.5 |
| Small ReLU | 25% | 99% | 479 | 5.5 |
| Medium ReLU | 3% | 100% | 491 | 4.92 |
| Tiny SoftPlus | 91 % | 100% | 238.33 | 7.4 |
| Small SoftPlus | 0% | 99% | 500 | 8.5 |
| Medium SoftPlus | 0% | 98% | 500 | 9.8 |

Table 4: Proportion of bound computations on CIFAR-10 where the algorithm converges within the iteration budget, and average number of iterations for each first order algorithm for an IBP trained network.

(a) Evolution of the duality gap as a function of time or number of iteration, for the NonConvex and DeepVerify Solver.

(b) Distribution of distance to potentially degenerate point.

Figure 7: Evaluation on the Medium-sized network with SoftPlus activation function trained with IBP.