[Reviews · NeurIPS 2020]

Review 1

Summary and Contributions: This paper focuses on a structured convex optimization problem which has constraints given in a stage-wise manner. The paper studies a nonconvex reformulation for this problem and proposes new algorithms to ensure convergence to global minimizers in degenerate cases. The paper shows that projected gradient descent (PGD) converges with a non-degeneracy assumption with a rate 1/epsilon. For the degenerate case, the authors first design a method to escape exact local minimizers. Then, the authors design a method to escape the basin of local minimizers and show that it takes epsilon^{-3} computations of the gradient to do so. Lastly. some experimental results are given which shows promising performance for the new formulation and the method.

Strengths: The stage-wise structure considered in the paper seems to have applications in the field, for which the authors have explanations in the supplementary material. I also liked the idea of exploiting this structure to define \phi_0 in a recursive manner. The new reformulation seems intuitive and and very suitable for the problem. In addition, the authors considered a host of different assumptions, and show algorithmic results for both non-degenerate and degenerate cases. Moreover, practical considerations such as the per-iteration cost or the number of gradient computations to escape local minimizers are also considered and discussed in the paper. Given the good practical performance, I believe the new reformulation can be quite promising for this class of problems. Moreover, this work can also motivate different reformulations by exploiting the structure on different problems.

Weaknesses: I have mostly minor comments about the weaknesses of the paper, mostly due to clarity. To me, these reasons are important to judge the significance of the contribution. I am willing to increase my score when these points are addresses. In section 1.2, the authors argue that the methods such as PDHG have worst-case behavior for the problem considered in this paper. First, it is not clear why the authors give a reference to Nesterov's book for this fact which concerns unconstrained minimization. Second, can the authors give the exact result they refer to in reference [16]? Moreover, as the authors also remark, the new method can also have very large constants in the bound (for example Appendix D or the number of computations is epsilon^{-3} in Thm 3 which can be quite pessimistic). Then, it is not clear if it will be better than PDHG, at least in theory, is this correct? Next, a practical comparison with PDHG would be suitable. I think it is also interesting to consider some toy examples to show how PDHG exhibits worst-case performance whereas the new method does not (if that is the case)? Right now, this comparison is not clear to me. If the new method has conservative theoretical properties but much better practical performance, I think then it would be nice for the authors to reflect on this and propose some open questions. This would show the strength of the new formulation. In such a case, what could be the reason for the unexpectedly (compared with theory) good behavior of the method in practice? Is it because the degenerate cases do not arise in practice? - The authors say that they use momentum and backtracking for the new method in practice. Are these variants straightforward to analyze using the results proven in this paper? If so, I think it would be nice to at least give some guidance how these results can be proven. If not, the reasons complicating the analysis should be discussed.

Correctness: Both the parts I checked in the proof and experimental methodology seem correct to me.

Clarity: I think that the paper is well-written. The sections have clear focuses (degeneracy, exact local minima, basin of local minimizers). Moreover, the remarks after the results are useful for the reader. As I detailed in Weaknesses part, more explanations can be added in some parts.

Relation to Prior Work: I think this part is sufficient but it can be extended. In particular, the discussion with first order methods need more elaboration (please see my specific comments in Weaknesses section.)

Reproducibility: Yes

Additional Feedback: ===== Update after rebuttal ==== After reading the rebuttal, I am happy with the answers regarding PDHG, and explanations about the proof for backtracking/momentum variants of the method. I find the approach promising. However, I think the work needs improvements in terms of presentation, which, I suggest the authors to consider when revising the paper. Especially the introduction of the idea can be made more friendly for the readers, with more explanations. I suggest the authors to also consider the minor questions in my review that are not mentioned in the rebuttal. ===== Update after rebuttal ==== - I find the way that the authors use \nabla sign a bit non-standard. I think the authors omit the argument of the gradient (as in \nabla f(x)) for brevity. It seems they rather use \nabla_x f to say the same thing. Can the authors clarify and explain this notation in the beginning of the paper? (I see the authors mentioning using f instead of f(x) but it would be nice to say that this also applies to gradients) - More explanation for Section 1.3 can be more useful for the reader. For example, why is \hat z_i defined, it seems like the authors use z_i = \hat z_i in the discussion here. Moreover, in eq. (5a) is it supposed to be a product, instead of addition in RHS? It seems in eq. (5a, 5b) the authors also use z_i = \hat z_i, is this accurate? - In line 110, definition of K_\gamma(x, \theta), I did not see the notation ()^+ and ()^- being defined. Also, can the authors spend more time explaining this definition. - line 127, regarding the variable d, the text under "minimize" is a bit convoluted, which takes some time to understand what the authors try to do, which is in fact a standard calculation. can the authors explain more to make it easier for the reader? - Algorithm 1, the name of the function is wrong. It seems to be that it should be SIMPLE_PSI_MINIMIZATION. - Remark 2: In the second sentence, in which sense the authors use the term "smooth"? Judging that Lipschitzness of the gradient is mentioned in the third sentence, I am not sure the authors mean L-smooth in the second sentence. - In terms of reproducibility, even though detailed proofs are given for the theoretical results, the code for the experiments is not given.


Review 2

Summary and Contributions: The authors propose to rewrite a special stage-wise optimization problem, which has applications in adverserial certification, into a more natural form. They replace the variable-dependent box constraint in each stage with a linear interpolation variable, which reduce the variable dependency between stages. With the reformulation, they applied PGD on the reformulated problem and try to bound the iteration of convergence under some non-crossing assumption. Further, they extend the algorithm to the case without a non-crossing assumption by adding a step to escape the local minimizer.

Strengths: The proposed method runs much faster than DeepVerify, with a slight improvement over the bound. Further, the authors claimed that PGD on the reformulate non-convex problem converges to a global minimizer and provided iteration bounds.

Weaknesses: There is some core parts in the proof I don't understand / cannot verify, so I don't know whether the proof is correct. My question is why PGD would produce a sufficient decrease in the highly non-convex optimization problem w.r.t. \psi. That is, I don't understand why in appendix C.3, (f(k) -f*)^2 <= const * (f(k)-f(k+1)). The property of the succeeding iterate (s^k+1, \theta^k+1) was not discussed in the proof.

Correctness: I verified the proof on Lemma 1 and do not find a mistake. However, I'm not sure the claimed global optimal convergence is correct due to my concern mentioned above.

Clarity: It's welll-written but a bit too dense. I suggest a subsection on the notations.

Relation to Prior Work: The authors only discuss limited related works in section 1.2.

Reproducibility: No

Additional Feedback: (Appendix line 16b) I think it is \partial s instead of \partial z. Also, I don't understand why you use \nabla f instead of \nabla \psi in (16a). ==== After rebuttal ==== The authors clarified the proof of the sufficient decrease in the rebuttal. Thus, I increase my score from 5 to 6.


Review 3

Summary and Contributions: This work reformulates a class of stage-wise convex optimization programs to certain nonconvex programs. The work establishes the equivalence of global minima between the two formulations. It proposes that the nonconvex formulation can be more efficiently solved using projected gradient descent than the original convex formulation. It studies conditions under which one can ensure global optimality of the proposed nonconvex algorithm(s).

Strengths: The work shows a variety of large-scale structured convex optimization problems can be reduced to a stage-wise form, and shows that such structures can be harnessed for efficient algorithmic development.

Weaknesses: The proposed algorithm(s) is essentially projected gradient descent type. However, to ensure that the algorithm will escape from local minimizers, various technical assumptions are made and special techniques are added to the process. The analysis is rather complicated and it is very difficult to evaluate how realistic those assumptions are and how some of the conditions are to be verified.

Correctness: The overall approach seems to be correct but it is almost impossible to check the correctness of all the details for the time given for a conference review. It is more proper for a related journal.

Clarity: The writing of the paper can be improved significantly, especially the organization of the technical assumptions, analysis, as well as justifications why the assumptions hold for the class of problems considered. The readability is a serious concern: given the amount of technical details, the current writing is certainly not suitable for a quick smooth read.

Relation to Prior Work: The proposed method, if claims are true, indeed improves previous convex approach to solve these class of specially-structured problems.

Reproducibility: Yes

Additional Feedback: The original problems are convex in nature. Could there be other natural convex re-formulation than can effectively harness the structures and yet lead to much more efficient algorithms? The nonconvex reformulation is somewhat unconventional and leads to complications in the analysis for ensuring global optimality. Escaping local minimizers for nonconvex problems are in general difficult... Maybe the authors want to elucidate more what make it possible for this class of problems? The analysis given in Appendix D does not give good intuition or explanation on this.


Review 4

Summary and Contributions: The paper proposes a novel nonconvex reformulation of stagewise convex problems that is amenable to efficient projected gradient descent. This is conceptually similar to nonconvex reformulations of SDPs. While a global solution of the convex problem is possible in theory, classical convex solvers are highly inefficient for these class of problems. The key idea is to eliminate/replace the optimization variables coupled via recursive constraints and treat them like the activations in deep net training. In fact, backbrop is used to compute the gradient of the reformulation and a forward pass is used to obtain the current value of the eliminated variables. As a main theoretical contribution the authors prove that non-spurious local minima correspond to global optima of the convex relaxation. The approach is applied to a convex relaxation of the problem of deep network verification, showing significant improvements (in terms of speed) over the deep verify [6] approach.

Strengths: The proposed approach seems to be innovative and the experimental results show significant improvements in terms of runtime over the baseline approach deep verify [6]. The reformulation of the problem seems to be sound.

Weaknesses: While the approach is novel and I basically like the ideas presented in the paper, I believe the clarity and readability of the paper should be improved. See below. I am also not able to verify the correctness of the very technical and long proof of the main result, but I believe this is in general a problem for this type of papers, that may therefore also fit an optimisation journal as an alternative venue, see below.

Correctness: I have the impression that the claims are correct, even though I cannot verify the correctness of the proofs provided. The empirical evaluation seems to be sound.

Clarity: The paper is well written in general, however, it is a bit tough to read. While for the the technical part, probably, not much could be done there, I believe for the introduction and the nonconvex reformulation part the authors may spend more effort to motivate and explain the problem clearly also providing more explicit examples. Suggestions below.

Relation to Prior Work: Besides the conseptual similarities to nonconvex reformulations SDP there is not much related work pointed out here. The authors may eleborate a bit more on the related work. Since deep net verification is the main application I suggest to at least elaborate on related work on the application side deep verify and other approaches for deep net verification.

Reproducibility: Yes

Additional Feedback: I believe the nonconvex reformulation could be motivated more clearly. I suggest to provide more examples, also the toy example (2) would help for that purpose. Maybe you can reference it earlier. I suggest to move some part of the supplements, e.g. the reformulation and convex relaxation of deep network verification A2 to the introduction, to make the main application and the precise reformulation very clear. There, I believe, in 9c the bias-term is missing. For the relaxation of the constraint \mu_i \leq z_i \leq \nu_i, it would help to just say in one sentence that this is equivalent to the existence of \theta_i \in [0,1] s.t. z_i = \theta \mu_i + (1-\theta) \nu_i, which is an additional optimization variable. It would also help if you could expand the recursive definition of \Psi_0 for n=3 to really see how the composition of functions actually looks like. I am also happy with the rebuttal as it clarified some points. I therefore stick to my accept vote.

[Author Response · NeurIPS 2020]

We thank the reviewers for their detailed feedback. We are encouraged by the fact that the reviewers appreciated the practical runtime improvements (**R1**, **R2**, **R4**), the theoretical contributions showing the soundness of our method (**R1**, **R3**, **R4**) and the appropriateness of our method (**R1** "very suitable for the problem", **R2** "a more natural form"). Most reviewers (**R1**, **R2**, **R4**) found the paper to be well-written. We provide clarification of specific concerns raised, which we will incorporate into the final version, along with other suggestions made in the reviews.

**R1 - Justification of the claims of worst case behaviour for PDHG/Toy example** Yes, [15] only provides lower bounds for unconstrained optimization but these techniques readily extend to constrained optimization. This exactly the idea that is used in [16, Section 3.1]. Replicating the argument presented in Section 3.1 for (2) establishes that (2) takes $n-1$ iterations for any saddle point algorithm. We admit that this might not be straightforward for readers not familiar with saddle point algorithm and lower bounds for convex optimization. Consequently, we will add a brief explanation on why [16, Section 3.1] can be used to establish that (2) takes $n-1$ iterations in the Appendix to the final version.

**R1 - It is not clear if it will be better than PDHG in theory, is this correct?** Yes. Although the constants on the two bounds are not directly comparable because they involve different quantities. It would be interesting in the future to see if better worst-case bounds are achievable, maybe with slightly stronger assumptions. As Remark 3 mentions PDHG provably performs poorly on (2) whereas our method solves the problem in one iteration.

**R1 - What is the reason for the much better practical permformance? Is it because the degenerate case does not arise in practice?** Indeed the one reason is that degenerate points do not appear in practice, see line 244-260 and Figure 4b. Also see lines 168-173 for some brief intuition for why does the method perform well in practice. The practical performance of this approach is not something we fully understand yet and we will add it as an open problem.

**R1 - Practical comparison to PDHG** We ran a test with both internal PDHG and mirror-prox code on a few problems and neither got close to optimal after 100,000 iterations. For the final version we will report results with (`https://odlgroup.github.io/odl/math/solvers/nonsmooth/pdhg.html`). Also, note in the paper we compare with SCS, which can be viewed as preconditioned PDHG (`https://arxiv.org/pdf/1811.08937.pdf`).

**R1 - Can the analysis be extended to the variant with backtracking and momentum?** Backtracking line search automatically works as it satisfies line 10 of Algorithm 1. The theory also applies to safeguarded momentum schemes. We will elaborate further on these points in the final version.

**R2 - I don't understand why in appendix C.3,** $(f(s^k, z^k) - f_*)^2 \leq \zeta_2(f(s^k, z^k) - f(s^{k+1}, z^{k+1}))$**.** We will update the reasoning in line 505-506 as follows. Define $\zeta_1 = 1$, $\zeta_2 = L\left(D_s\sqrt{2} + 2\frac{c + D_s\|W\|_2 + D_z\|V\|_2}{\gamma}\right)^2$, and $\zeta_3 = \frac{2\Delta(s^1, \theta^1)}{L}$. Lemma 1 shows that if $\delta_L(s^k, \theta^k) \leq \Delta(s^1, \theta^1)/\zeta_3$ then $\Delta(s^k, \theta^k)^2 \leq \zeta_2\delta_L(s^k, \theta^k)$. Furthermore, $\delta_L(s^k, \theta^k) \leq \psi_0(s^k, \theta^k) - \psi_0(s^{k+1}, \theta^{k+1}) = f(s^k, z^k) - f(s^{k+1}, z^{k+1})$ by line 10 of Algorithm 1 and the definition of $\psi_0$ respectively. Combining these two inequalities yields $(f(s^k, z^k) - f_*)^2 \leq \zeta_2(f(s^k, z^k) - f(s^{k+1}, z^{k+1}))$. Applying Lemma 1 to the latter inequality yields the result.

**R3 - It is difficult to evaluate how realistic the assumptions are and how some of the conditions are to be verified** Assumption 1 and 2 are standard assumptions (smoothness and bounded feasible region) that are usually easy to verify. In Appendix A, we verify assumption 3 for all the examples presented.

**R3 - Escaping local minimizers for nonconvex problem is in general difficult ... maybe the authors want to eludicate more on what makes it possible for this class of problems?** Section 3.1 proves that we can escape local minimizers in the exact case and therefore is a good source of intuition for the inexact case (Section 3.2 and Appendix D). Also, see line 109-124 and Figure 2. The final version will expand on these.

**R3 - Would there be convex reformulations that could similarly harness the structure but not introduce the complications of non-convexity?** Despite this problem being well studied, we are not aware of any such convex reformulations. We sacrifice convexity to replace 'hard' convex constraints with easy bound constraints. We suspect there is no convex reformulation that achieves this property. Similarly, nonconvex reformulations of SDPs sacrifice convexity to eliminate the constraint that $X$ is positive semidefinite. There are also no known convex reformulations of SDPs that eliminate the 'hard' SDP constraint.

**R4 - Elaborate more on the related work on Deep Net verification.** We will add the following text to the end of the introduction, before subsection 1.1. *The convex relaxation was proposed by Ehlers [Ehlers, ATVA 2017] but the high computational cost of solving it with off-the-shelf LP solvers was prohibitive for large instances. A large number of papers such as IBP [Gowal et al., ICCV 2018], DeepPoly[Singh et al, POPL 2019], Crown [Zhang et al., NeurIPS 2018], Neurify [Wang et al., NeurIPS 2018], LP-relaxed-dual [Wong & Kolter, ICML 2018] focused on looser relaxations to allow for fast, closed form solutions of the bound computation problem scaling to larger networks. In parallel, work was done to reformulate the optimization problem to allow the use of better algorithms: DeepVerify [Dvijotham et al., UAI 2018] introduced an unconstrained dual reformulation of the non-convex problem and showed equivalence with the convex relaxation. Proximal [Bunel et al., UAI 2020] performed lagrangian decomposition and used proximal methods to solve the problem faster. The application of our method to network certification follows this research direction of speeding up the computation of network bounds without compromising on tightness.*

[Meta-Review · NeurIPS 2020]

The paper considers structured convex optimization where constraints are given in a stage-wise manner. The paper studies a non-convex reformulation for this problem and proposes new algorithms to ensure convergence to global minimizers for both non-degenerate and degenerate cases. The reformulation is proven effective in theory and experiments. The author feedback phase has clarified several aspects, resulting in a consensus on weak acceptance. We hope the detailed feedback with improvement suggestions from the 4 reviews will be implemented for the camera ready version, in particular about the clarity and readability of the paper.